

# Deriving transmission losses in ephemeral rivers using satellite imagery and machine learning

Antoine Di Ciacca[1], Scott Wilson[1], Jasmine Kang[2], and Thomas Wöhling[1,3]

[1]Environmental Research, Lincoln Agritech Ltd, Lincoln, New Zealand
[2]National Institute of Water and Atmospheric Research (NIWA), Christchurch, New Zealand
[3]Chair of Hydrology, Technische Universität Dresden, Dresden, Germany

**Correspondence:** Antoine Di Ciacca (antoine.diciacca@lincolnagritech.co.nz)

**Abstract.** Transmission losses are the loss in the flow volume of a river as water moves downstream. These losses provide crucial ecosystem services, particularly in ephemeral and intermittent river systems. Transmission losses can be quantified at many scales using different measurement techniques. One of the most common methods is differential gauging of river flow at two locations. An alternative method for non-perennial rivers is to replace the downstream gauging location by visual assessments of the wetted river length on satellite images. We used this approach to estimate the transmission losses in the Selwyn River (Canterbury, New Zealand) using 147 satellite images collected between March 2020 and May 2021. The location of the river drying front was verified in the field on five occasions and seven differential gauging campaigns were conducted to ground-truth the losses estimated from the satellite images. The transmission loss point data obtained using the wetted river lengths and differential gauging campaigns were used to train an ensemble of random forest models to reconstruct the hourly time series of transmission losses and their uncertainties. Our results show that the Selwyn river transmission losses ranged between 0.25 and $0.65\,\mathrm{m}^3/\mathrm{s}/\mathrm{km}$ during most of the 1-year study period. However, shortly after a flood peak the losses could reach up to $1.5\,\mathrm{m}^3/\mathrm{s}/\mathrm{km}$. These results enabled us to improve our understanding of the Selwyn River groundwater – surface water interactions and provide valuable data to support water management. We argue that our framework can easily be adapted to other ephemeral rivers and to longer time series.

## 1 Introduction

Transmission losses are the loss in the flow volume of a river as water moves downstream (Walters, 1990). An important consideration of this definition is that transmission loss refers to all of the water lost by a river – evaporation, transpiration by macrophytes and riparian vegetation, as well as groundwater recharge (Mcmahon and Nathan, 2021). In dryland regions, where water scarcity is a major issue, rivers are often ephemeral or intermittent (i.e. non-perennial), and are thought to be the primary source of groundwater recharge (Shanafield and Cook, 2014; Wang et al., 2017). In addition, intermittent and ephemeral rivers shelter specific freshwater biodiversity and play an important role in biogeochemical cycles (Datry et al., 2014; Fovet et al., 2021). Interactions between non-perennial rivers and groundwater can be particularly complex with, for example, the development of perched aquifers during high flows (Shanafield et al., 2021; Villeneuve et al., 2015; Wheater et al., 2010).



The quantification of transmission losses and groundwater – surface water interactions has been approached in many different ways (Cook, 2015; Kalbus et al., 2006). The methods used to estimate transmission losses can be classified in three groups after Shanafield and Cook (2014), depending if they rely on measurements of streambed infiltration, groundwater state variable(s) or river discharge. Estimating the streambed infiltration typically gives point estimates and can be done directly with seepage meters (e.g. Lee, 1977; Lee and Cherry, 1979; Rosenberry et al., 2020) or indirectly using tracers (e.g. González-Pinzón et al., 2015; Hatch et al., 2006; Le Lay et al., 2019). However, as stated by Cook (2015), small scale estimates cannot be easily extrapolated to larger scales relevant for water management. Another way to approach this problem is to conduct measurements in the groundwater in order to determine the river recharge response signal. This can provide larger scale estimates of the groundwater recharge by means of hydraulic (e.g. McDonald et al., 2013) or chemical measurements (e.g. Hoehn and Von Gunten, 1989; Massmann et al., 2009; Popp et al., 2021; Schaper et al., 2022). Unfortunately, these estimations are often complicated by the amount of information needed on the aquifer properties, which cannot be easily estimated at the appropriate scale. Finally, transmission losses can be quantified by differential gauging of river flow at two locations. Although river flow is routinely measured in many hydrological studies, these measurements are rather labour-intensive and it is difficult to record high flow events, which occur over very short periods. An easier way to generate river discharge time series is to monitor the river level and generate a stage-discharge rating curve to determine discharge. However, the use of a rating curve introduces uncertainties on the river discharge values, which can be considerable and are often underestimated (Di Baldassarre and Montanari, 2009; McMahon and Peel, 2019; McMillan et al., 2012).These uncertainties become even bigger when two river gauging stations are used to calculate the transmission losses, as the uncertainties are compounded. Furthermore, as noted by Cook (2015), it is unusual for two gauging stations to be located on the same river without intervening tributaries. Therefore, quantifying transmission losses from two existing gauging stations is rarely possible. For ephemeral rivers, an alternative approach using satellite observations has been introduced by Walter et al. (2012). In this approach, the length of the wetted reach downstream a flow gauging station is visually identified on satellite images. The transmission losses can then be calculated by dividing the river flow at the gauging station by the wetted river length. Walter et al. (2012) used this approach to calculate the transmission losses in the Frio River (Texas, United States) using five images collected between 1994 and 2008.

In combination with measurements, transmission losses and groundwater – surface water interactions can also be quantified using models (Fleckenstein et al., 2010; Lewandowski et al., 2020; Mcmahon and Nathan, 2021). A wide variety of models have been used for this purpose. Early attempts include a linear relationship between the flow rate and the river – aquifer head difference, based on a constant streambed resistance only (Prickett and Lonnquist, 1971). This relationship is still widely used nowadays as it is implemented in the popular MODFLOW family of codes (Harbaugh, 2005; Harbaugh et al., 2000; Langevin et al., 2017; McDonald and Harbaugh, 1988). However, numerous studies have suggested that this is an oversimplification of the system in many cases and some proposed alternative expressions (Anderson, 2005; Di Ciacca et al., 2019; Morel-Seytoux et al., 2018; Rupp et al., 2008; Rushton, 2007; Rushton and Tomlinson, 1979). Nevertheless, these alternative expressions rely themselves on numerous assumptions that make them often unsuitable to represent the complex interactions between groundwater and non-perennial rivers. Recently, fully coupled models have been developed with the aim of representing the interactions between groundwater and surface water in all their complexity (e.g. Fatichi et al., 2016; Kuffour et al., 2019;





Maxwell et al., 2009; Therrien and Sudicky, 2006). However, this complexity and the resultant data requirements make them difficult tools to use. Moreover, they need to be calibrated and evaluated on independent data in order to demonstrate their benefits over simpler solutions.

An alternative approach that has gained popularity in the hydrological modelling community over the last decades is machine learning (Shen et al., 2021; Solomatine and Ostfeld, 2008; Tran et al., 2021). These algorithms can be very efficient at
reproducing the response variable (e.g. transmission losses) with minimum user assumptions, providing that enough training data are available. A machine learning algorithm particularly capable of representing non-linear and complex relationships between variables is random forest. This approach builds an ensemble (a forest) of small decision trees for the response variable by subsampling the predictor data using random combinations of predictor variables. The results of the 'forest' are aggregated to determine the ensemble majority (classification) or average (regression) result for the response variable
(Breiman, 2001; James et al., 2013). Random forests have been successfully used in hydrogeology to predict the origin of samples, nitrate contamination and redox conditions in groundwater (Baudron et al., 2013; Knoll et al., 2019; Koch et al., 2019; Rodriguez-Galiano et al., 2014; Wilson et al., 2020). Despite being less common than other machine learning approaches (e.g. artificial neural network, support vector machines), random forests have also been used in hydrology, including for estimating various hydrological indices at ungauged sites and streamflow forecasting (Booker and Woods, 2014; Desai and
Ouarda, 2021; Papacharalampous and Tyralis, 2018; Tyralis et al., 2019).

In the coastal plains of New Zealand, most of the groundwater recharge is thought to be sourced from gravel-bed river water infiltration. For example, the annual land-recharge is only around 3% of the river recharge in the Heretaunga Plains (Dravid and Brown, 1997) and contributes to less than 4% of the water balance in the Wairau Aquifer (Wöhling et al., 2018). In the Central Plains of the Canterbury Region, the Waimakariri River is providing more than 80% of the spring-fed Avon river
baseflow and is the major source of groundwater for the Christchurch City area (White, 2009; White et al., 2012). In these regions, groundwater resources are under increasing pressure to meet the demand for municipal, agricultural, and industrial uses (Brown et al., 1999; Rosen and White, 2001; Smith and Montgomery, 2004; Wöhling et al., 2020). The most important rivers for groundwater recharge in New Zealand have often a high braiding intensity, with several channels resulting in wide braid plains (> 1 km). Interactions between braided rivers and groundwater have receive little attention so far, and the quantity
of water lost by these rivers and the main recharge mechanisms involved are still largely unknown. This makes any simulation of plausible future scenarios very delicate. Recently, Coluccio and Morgan (2019) published a review of methods for measuring groundwater – surface water exchange in braided rivers, highlighting the difficulties inherent to this kind of river. In the Central Plains of Canterbury, the Selwyn River has been previously used as a benchmark system for undammed alluvial rivers that are under intense pressure for water abstraction (Arscott et al., 2010; Datry et al., 2007; Larned et al., 2011, 2010, 2008, 2007;
Rupp et al., 2008). Its relatively small width and low braiding intensity (1-2 channels most of the time), allow for an easier instrumentation and investigation than larger braided rivers. Furthermore, the Selwyn River includes an ephemeral losing reach, for which we could derive an extensive dataset of transmission losses using satellite imagery.

In this article, we present a framework to estimate transmission losses from satellite imagery and reconstruct their time series using random forest regressors. To estimate the transmission losses using satellite imagery, we used a similar approach





than Walter et al. (2012) but on a more comprehensive library, with different image sources and with field data to verify our estimations. We then used the transmission loss point data obtained to train an ensemble of random forest models. This ensemble enable us to reconstruct the hourly time series of transmission losses and their uncertainties. This constitutes another novelty of our approach.

The paper is organized as follows. Section 2 presents our study site on the Selwyn River (New Zealand). Next, section 3

details the methods adopted to gauge the river flow, estimate the river transmission losses and reconstruct the hourly time series. In section 4, first the results of one flood event are described, second our complete dataset is analysed and third the reconstructed hourly time series are presented. Finally, section 5 discusses the physical interpretation, the advantages and limitations of our approach and outlines possible future developments and applications, before section 6 concludes with a summary of the most important findings.

## 2 Study Site

The Selwyn River flows on 93 km from the foothills of the New Zealand Southern Alps across the alluvial Central Plains of the Canterbury Region to the Lake Ellesmere and the Pacific Ocean (Figure 1). The river course mainly follows a depression between the alluvial fans of the much larger Waimakariri and Rakaia Rivers. In the foothills, the Selwyn River is constrained by hillslopes and has a meandering single-thread channel. This constrained reach is perennial and gaining water from the

surrounding hills. When it arrives in the inland plains, the Selwyn channel slope decreases abruptly and it becomes braided or semi-braided. The $3 \, \text{km}$ long perennial reach loses water to the underlying aquifer due to the thickening of the gravel assemblage as the river leaves the confines of the foothills. As the transmission loss increases, the river becomes ephemeral for around $30 \, \text{km}$ of its length. Further downstream towards the coastal plain, the Selwyn River gains water from groundwater seepage, and becomes first intermittent and then perennial again as the coast is approached (Larned et al., 2008; Rupp et al.,

115 2008).

In this study, we focus on the first part of the ephemeral losing reach, extending for $15 \, \text{km}$ upstream of the confluence with the Waianiwaniwa and Hororata Rivers (Figure 1). The studied reach flows through the inland plains, which are dominated by glaciofluvial gravels covering Cretaceous and Tertiary sedimentary basement rock to depths of $120 - 160 \, \text{m}$ (Taylor et al., 1989; Wilson, 1973). In this region, aquifers are complexes of interbedded gravels, partially separated by leaky aquicludes.

The shallowest aquitards below the Selwyn River at the study area are $20 - 50 \, \text{km}$ deep (Vincent, 2005). Groundwater flows sub-parallel to the direction of the Selwyn River, following the topographic gradient and the anisotropic permeability in the aquifer gravels (Burden, 1984). Aquifers in this region are recharged by water leaking through river channels and infiltrating though the land surface. The Selwyn climate, geology, hydrology and geomorphology have been extensively described by Larned et al. (2008).



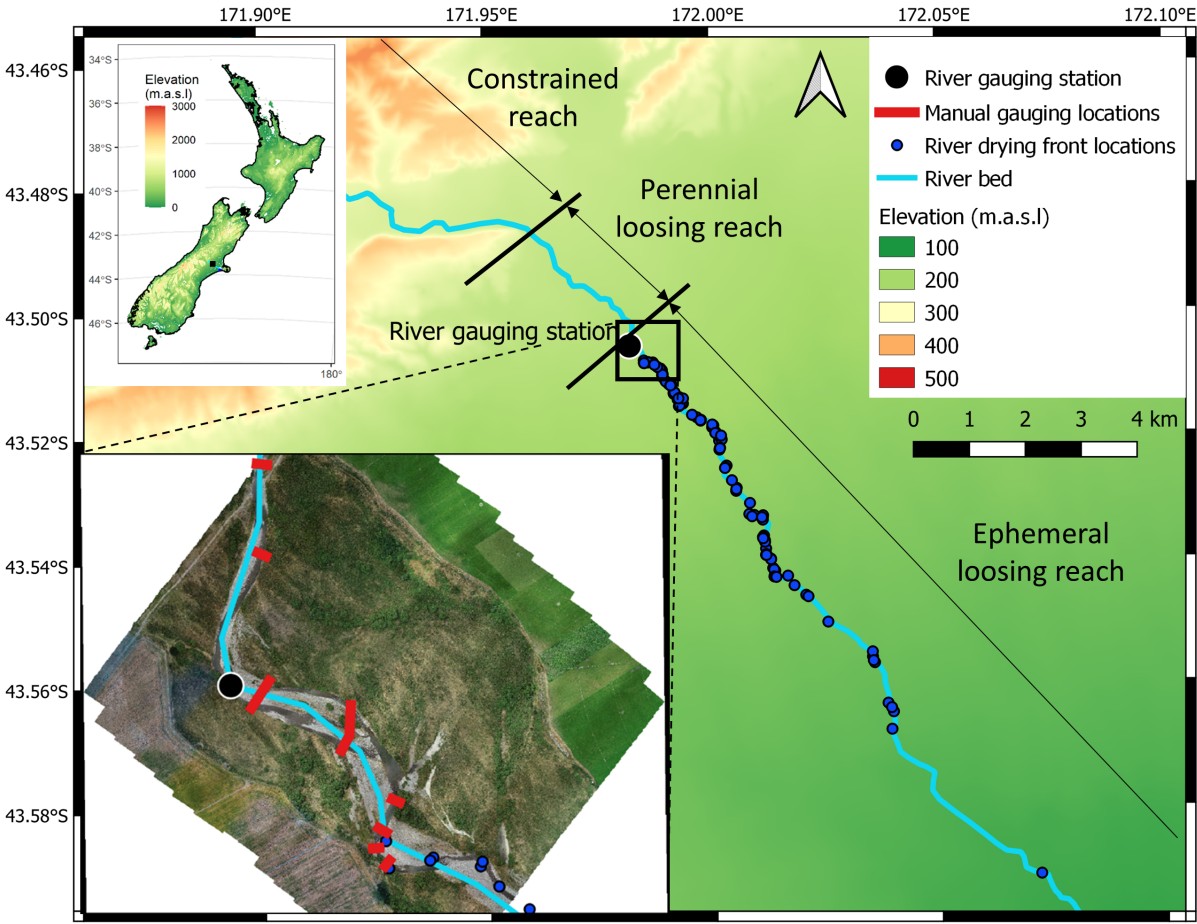

**Figure 1.** Map of the Selwyn River with the river gauging monitoring station, the 8 manual gauging cross sections and the 152 river drying front locations. The different reaches were delimited according to Larned et al. (2008).

## 3 Methods

### 3.1 River discharge time series

#### 3.1.1 River discharge measurement

The river discharge time series was derived from the river stage, monitored continuously at a stable cross-section, and a stage-discharge rating curve that relates the discharge to the recorded stage. The river stage was monitored at the upstream boundary of the ephemeral losing reach (referred to as 'Scotts Road', Figure 1), for the period from March 2020 to May 2021, with a Seametrics PT12 pressure transducer (5 m range). The typical accuracy of this sensor is 2.5 mm and the associated uncertainties were propagated to the rated flows. The river stage is reported as a height of water above a local datum around 209 m.a.s.l..





The stage-discharge rating curve of the Selwyn River at Scotts Road was developed using 14 manual flow measurements collected from April 2020 to March 2021 using either a SonTech Flowtracker or an Acoustic Doppler Current Meter (ADCP,
RDI StreamPro). These manual discharge measurements ranged from $0.22$ to $10.12\,\mathrm{m^3/s}$ and were conducted when the river stage was between $0.86$ and $1.3\,\mathrm{m}$. The uncertainties of the manual gauging data varied from 2.4 to 6.5%. The rating curve was defined using five inflection points and the fitting errors between our manual flow measurements and the rating curve ranged from 0 to 15%, with an average of 5 and a standard deviation of 7%. Considering the flow measurement uncertainties and the rating curve fitting errors, we assumed 20% of uncertainty on the rated river flows. The rating curve is presented in Appendix
A together with the measurement and inflection points used to develop it and the associated uncertainties.

### 3.1.2  Hydrograph processing

The hydrograph obtained from the stage record and the rating curve was processed in order to extract the peaks higher than $0.3\,\mathrm{m^3/s}$. First, we have identified each peak by automatically finding the time at which the first order derivation became negative. They were then filtered using an iterative procedure to only select the peaks higher than $0.3\,\mathrm{m^3/s}$ and no more than
one peak per $48\,\mathrm{h}$. The peak height was taken as the difference between the peak flow value and the minimum before the peak. These peaks were used to calculate the time since the last peak and the peak height associated with each transmission loss estimate. On one occurrence (April 3, 2021), the satellite image was taken just before ($3\,\mathrm{h}$) the peak flow was reached, we therefore used the time to the peak instead of the time since the peak. The time since the last peak and the peak height were used to understand and predict the transmission loss dynamics. The hydrograph and the selected peaks are presented in
Appendix B.

### 3.2  Estimation of the river transmission losses

We have estimated the Selwyn River transmission losses following two different approaches. The first is a similar approach than adopted by Walter et al. (2012) but using a much more comprehensive library of satellite images and constitutes the originality of this study. It uses the drying front location of the ephemeral reach to determine the wetted river length. The second is a more
traditional differential gauging approach and is used as a comparison on several days.

### 3.2.1  Transmission losses derived from the river drying front locations

The average river transmission losses along the reach downstream of our gauging station ($q_{\mathrm{loss}}$, [L$^2$.T$^{-1}$]) were calculated by dividing the river discharge ($Q$, [L$^3$.T$^{-1}$]) by the wetted river length ($L$, [L]):

$$q_{loss} = \frac{Q}{L} \tag{1}$$

L was estimated by measuring the wetted river length from the gauging station to the river drying front location.

The Selwyn River drying front was located on 147 satellite images taken between April 2020 and May 2021. We used satellite images available in the Planet Monitoring collection, which are mainly taken by the Dove satellite constellation and provide $3.7\,\mathrm{m}$ resolution images of the entire Earth daily in four multispectral bands: RGB (Red, Green, Blue) and Near





Infrared (Planet Team, 2017). Additionally, the drying front location was verified on the field on five different days in March 2021 using a GPS (Global Positioning System) device (Trimble R10 with a centimetre-level accuracy). The location of the 152 drying fronts along the riverbed are presented in Figure 1.

We considered two sources of uncertainty on the wetted river length estimation. The first one is related to the difficulty to identify accurately the drying front location on the satellite images. A comparison between the GPS and satellite drying front positions showed us that this uncertainty could be up to $100\,\text{m}$. The second source of uncertainty is the determination of the distance between the drying front and the gauging station (i.e. wetted river length). The wetted river length can differ depending if the river active channel, the riverbed or the braid plain is followed. We adopted the intermediate option of following the riverbed but assumed 10% uncertainties on the wetted river length estimations to account for this vague definition. The transmission loss estimates derived from satellite images include these two sources of uncertainty while the estimates made using the GPS points include only the second one.

### 3.2.2 Transmission losses derived from differential flow gauging

We conducted seven differential flow gauging surveys close to the upstream boundary of the ephemeral losing reach. During each survey, the river flow was measured at eight cross sections. Some cross sections included multiple braids; this resulted in 12 gauging locations along a river reach of $700\,\text{m}$, covering three riffle-pool sequences (Figure 1). The uncertainties of these manual flow measurements depend on instrument and site constraints. For our measurements, the uncertainties were estimated between 2.7 and 6.3%; the higher uncertainties are typically associated with shallow and low flow in the smaller braids.

The reach scale average transmission losses were calculated by fitting linear models to the relationships between the river discharge and the distance from the first upstream gauging location. The transmission loss values are the slope of the linear models. To transfer the measurement uncertainties to the transmission loss estimates, we have fitted a linear model to each of 10,000 realizations sampled in the uniform distributions representing the measurement uncertain ranges. The flow gauging measurements, their uncertainties and the linear model ensembles are presented in Appendix C.

### 3.3 Time series reconstruction using random forest regression models

Random forest regression models were trained on the dataset to reconstruct the transmission loss time series with an hourly timestep. We used the 'tidymodels' framework implemented in the R language (Kuhn and Wickham, 2020) and the 'ranger' implementation of random forest (Wright and Ziegler, 2017) with 1,000 trees per forest. The random forests were trained with three predictor variables, the river stage, the time since the last peak (log10 transformed) and the height of this peak. In the course of the model development, more predictors (e.g. river flow, water temperature, groundwater level, date) have been tested but they appeared to not improve significantly the predictions. We have selected the model with the lowest dimension among the better performing ones. The randomly selected predictor number was set to two and the minimal node size to one. We used 75% of the data to train the models and kept the other 25% for testing them. A stratified sampling was applied to ensure that the distribution of the time since the last peak was similar in the training and testing datasets. For more clarity, we refer to the





transmission losses predicted by the random forest models as 'reconstructed', as opposed to the 'estimated' values from the field data and satellite images.

To propagate the uncertainties of our estimated transmission losses through the reconstruction, we trained a random forest on each of 10,000 realizations sampled in the uniform distributions representing the estimated uncertain ranges. For each

realization, a different training and testing datasets were selected. Thus, we obtained an ensemble of random forests that we used to represent the uncertainties on the reconstructed values. The use of random forests is advantageous in this case because they are computationally fast, particularly when implemented in ranger which is also memory efficient (Wright and Ziegler, 2017). This efficiency enables an ensemble to be generated for the purpose of describing uncertainties, an approach that would be difficult with other machine learning methods that are more computationally demanding.

Given the stochastic nature of our estimation and reconstruction, the evaluation of the random forest fits against the estimated losses gives us multiple residual values for each estimation. We report hereafter the average root mean square error (RMSE) and the average normalized RMSE (NRMSE, normalized by the mean) of the 10,000 realizations. These evaluation metrics assess how well the random forest realizations could fit the training and testing data points, sampled in the uniform distributions representing the estimated uncertain ranges. Furthermore, to evaluate the ability of our ensemble to reproduce the estimated

transmission losses, we report the RMSE and NRMSE of the average predicted versus average estimated values.

Lastly, we computed a transmission loss duration curve by calculating the exceedance probability of the reconstructed hourly values in the same approach for generating flow duration curves. This analysis was done considering one year of data from May 1, 2020 to May 1, 2021.

## 4   Results

The estimated transmission losses vary in time with a complex relationship to the river stage and discharge, depending on the timing after flood events. We first explain this relationship for one particular event in September 2020, then show the complete dataset of estimated values and lastly present our reconstructed time series.

### 4.1   September 2020 flood event

The flood event occurring on September 18, 2020 was selected for explaining the transmission losses behaviour because the

satellite imagery coverage was particularly good. This allowed us to monitor the transmission losses time-dynamic during the first days after peak flow (Figure 2). Furthermore, a differential gauging field campaign was conducted on September 24, 2020; a day for which we also have a satellite image. This enables a comparison between the two approaches six days after peak flow and thus a verification of our method.

During this event, the peak flow was reached around 9 am on September 18. However, the wetted river length continued to

increase for around two days before it stabilized and started decreasing around three days after peak flow. Hereafter, we refer to the periods during which the wetted river length is either increasing or stable as 'wetting phases' and the periods during which the wetted river length is decreasing as 'drying phases'. For this event, transmission losses were maximum at peak flow, around



$1.1\,\mathrm{m^3/s/km}$. Then, they decreased linearly with the logarithm of the time since the last peak during the wetting phase (Figure 3). Finally, they stabilized around $0.35\,\mathrm{m^3/s/km}$, 3-4 days after the peak, during the drying phase. The transmission losses were non-linearly positively correlated with the river stage, with a relationship resembling a polynomial function (Figure 4). Furthermore, the transmission losses estimated for September 24, 2020 from differential gauging, and from the drying front location identified on a satellite image, correspond well given their respective uncertainties.



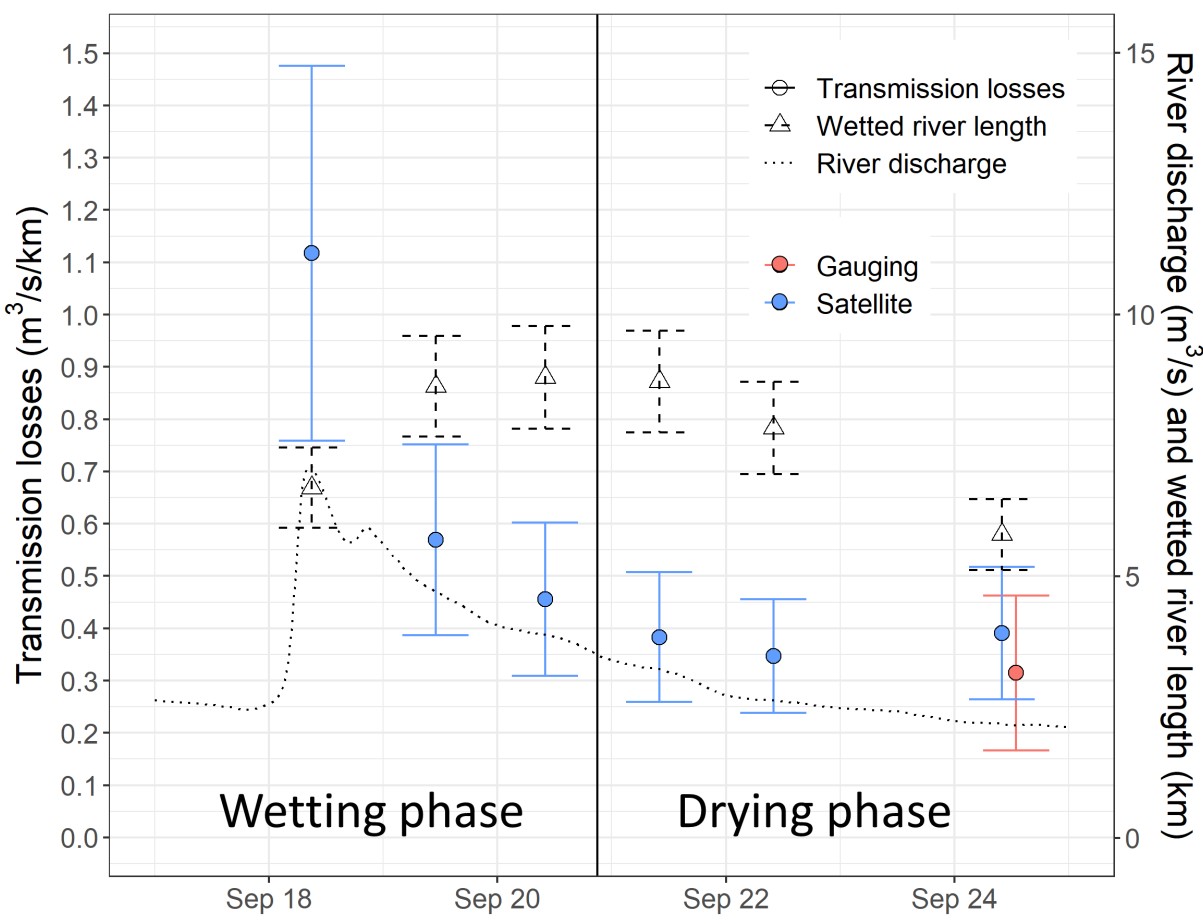

**Figure 2.** Time series of the river discharge (black dotted line), wetted river length (black dashed error bars and triangles) and transmission losses (solid error bars and circles) estimated using differential gauging ('Gauging', red) and river drying front locations identified on satellite imagery ('Satellite', blue) during the September 2020 selected event.




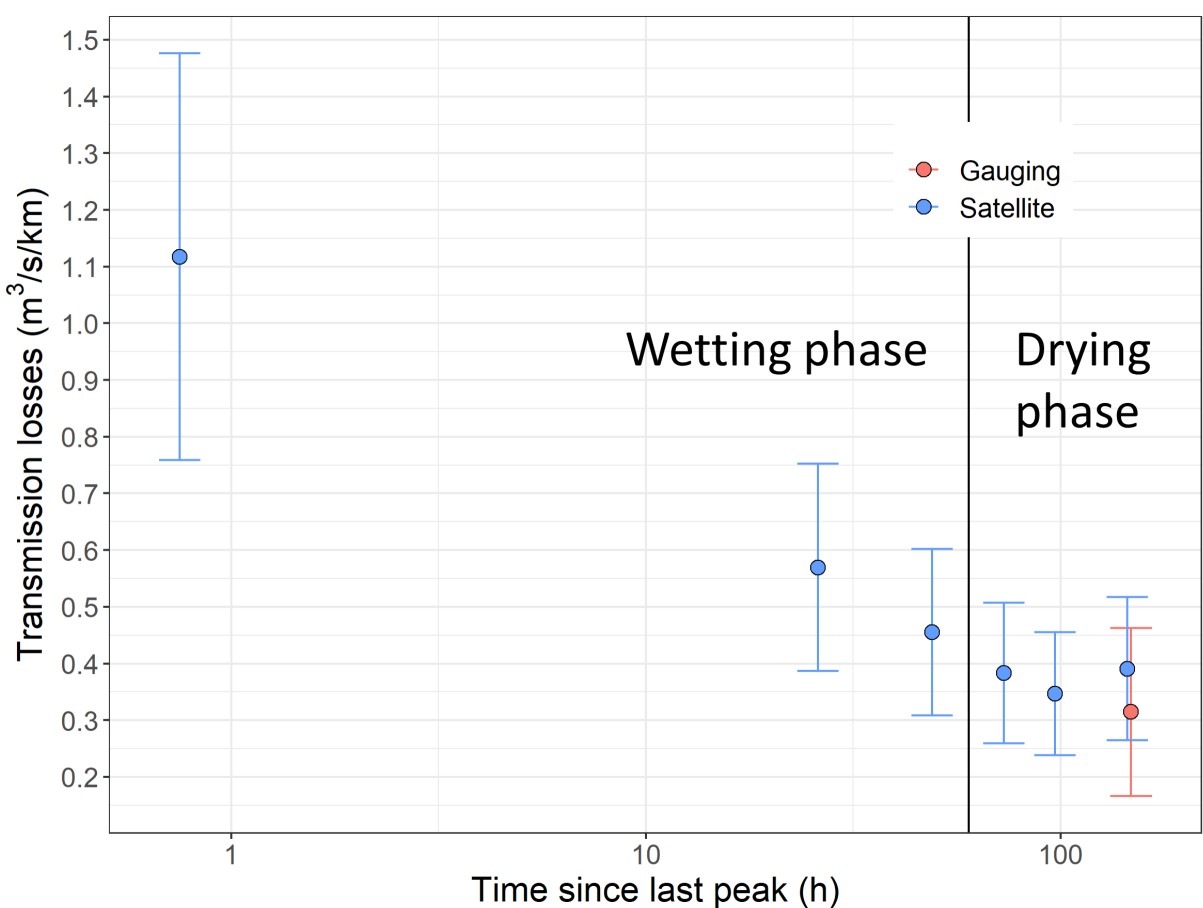

**Figure 3.** Selwyn river transmission losses as a function of the time since the last peak (log10 scale) during the September 2020 selected event.





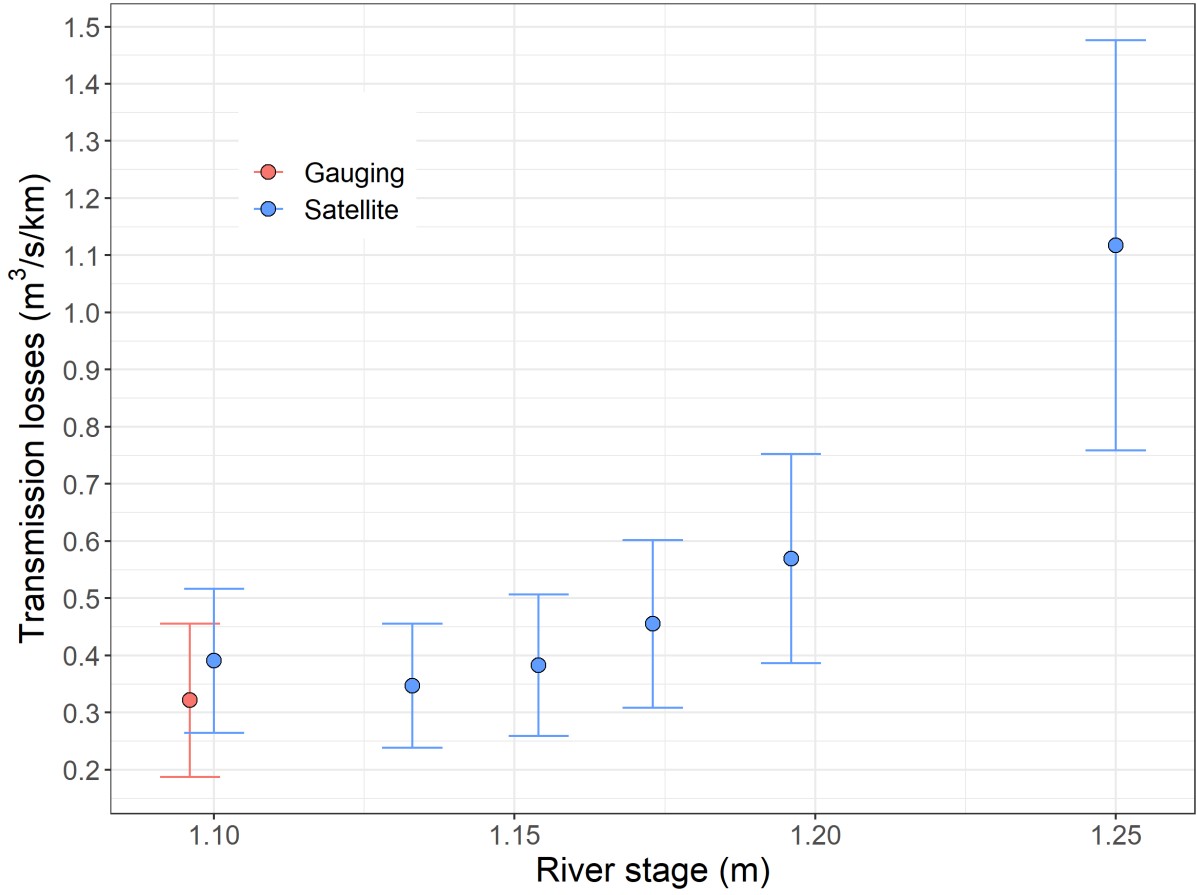

**Figure 4.** Selwyn river transmission losses as a function of the river stage during the September 2020 selected event.

## 4.2 Complete dataset of transmission losses

The transmission loss time series follow the pattern described in section 4.1 but for many more events of different magnitude

(Figure 5). The estimated transmission losses range from $0.14$ to $1.5\,\mathrm{m^3/s/km}$. Most of the estimated losses (56%) are below $0.6\,\mathrm{m^3/s/km}$ and correspond mainly to baseflow periods and river drying phases. The lowest values are found during dry periods, from March to May 2021, when the river stage and discharge were low. The highest losses occur shortly after high flow events, during wetting phases. Although it can be noted that the differential gauging estimates are lower in most instances, the transmission losses calculated with the different approaches correspond well given their respective uncertainties.

When the river stage and discharge became particularly low after April 2021, the river length downstream of our gauging station decreased to a few hundred meters. As a consequence, the uncertainties on our transmission loss estimates increased drastically. In the remaining of this article, we exclude the estimates for which the uncertainty is superior to 45% of their estimated value.





The relationship between the transmission losses and the river stage is presented in Figure 6. At low flow (up to $1\,\mathrm{m}$ stage
and $1\,\mathrm{m^3/s}$ discharge), the relationship between the river stage and the transmission losses is relatively linear and the estimated
transmission losses vary from $0.14$ to $0.83\,\mathrm{m^3/s/km}$. The effect of the small peaks is minor even for losses estimated shortly
after peak flows. At higher flow ($> 1\,\mathrm{m}$ stage and $1\,\mathrm{m^3/s}$ discharge), transmission losses stop increasing linearly with the river
stage and the effect of the peaks becomes an important control. The transmission losses respond to river stage and discharge
dynamics as explained in section 4.1. Flow losses decrease linearly with the logarithm of the time since the last peak during
wetting phases and the height of the peak appears to control the maximum values, which could reach more than $1\,\mathrm{m^3/s/km}$
in several instances (Figure 7). The relation between the transmission losses behaviour and hydrological processes is further
discussed in section 5.1.





**Figure 5.** Time series of the river discharge (black dotted line) and transmission losses (error bars) calculated using the differential gauging ('Gauging', red) and the river drying front method with field GPS measurements ('GPS', green) and satellite imagery ('Satellite', blue).

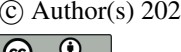



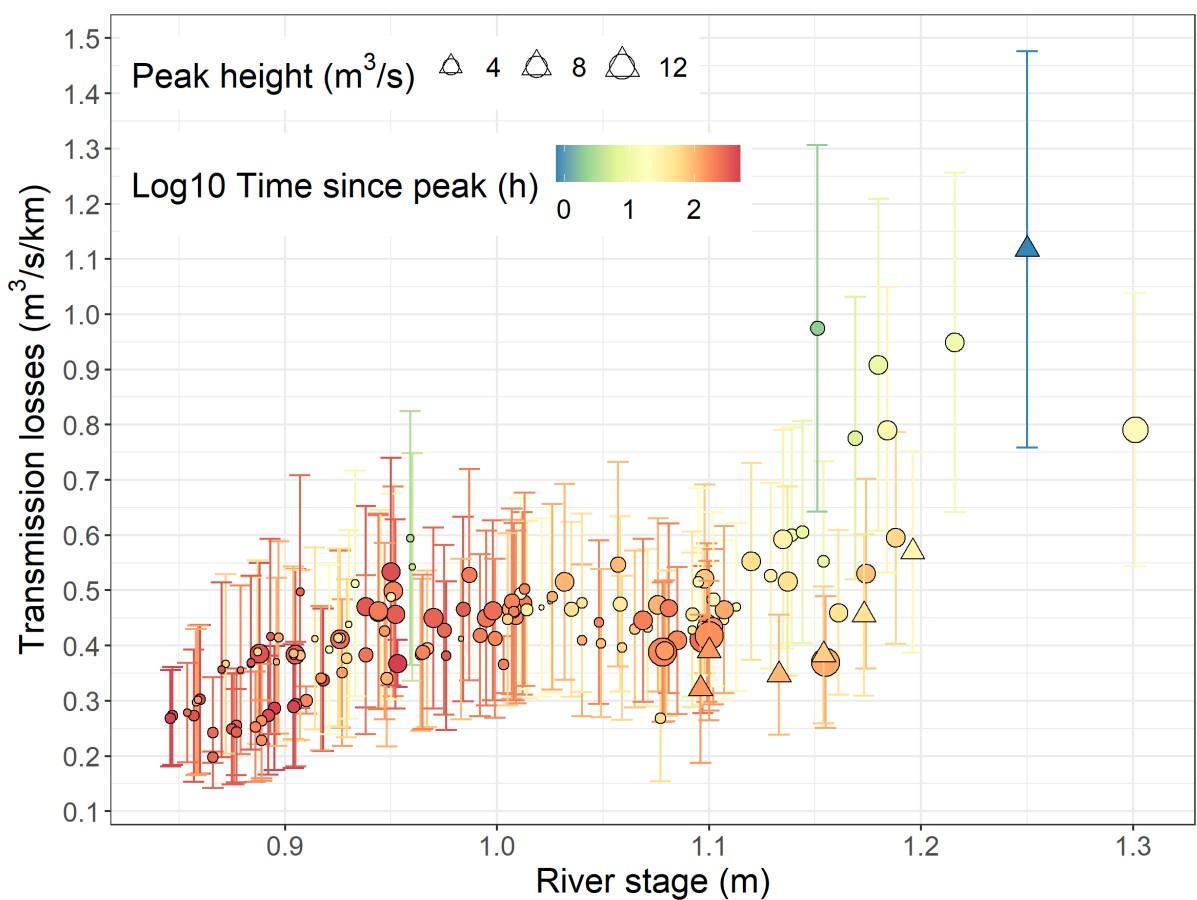

**Figure 6.** Estimated transmission losses as a function of the river stage. The colour scale represents the time since the last peak (log10 transformed) and the point size scale represents the peak height. Triangles indicate the September 2020 event presented in Figure 4 and circles the other data points.



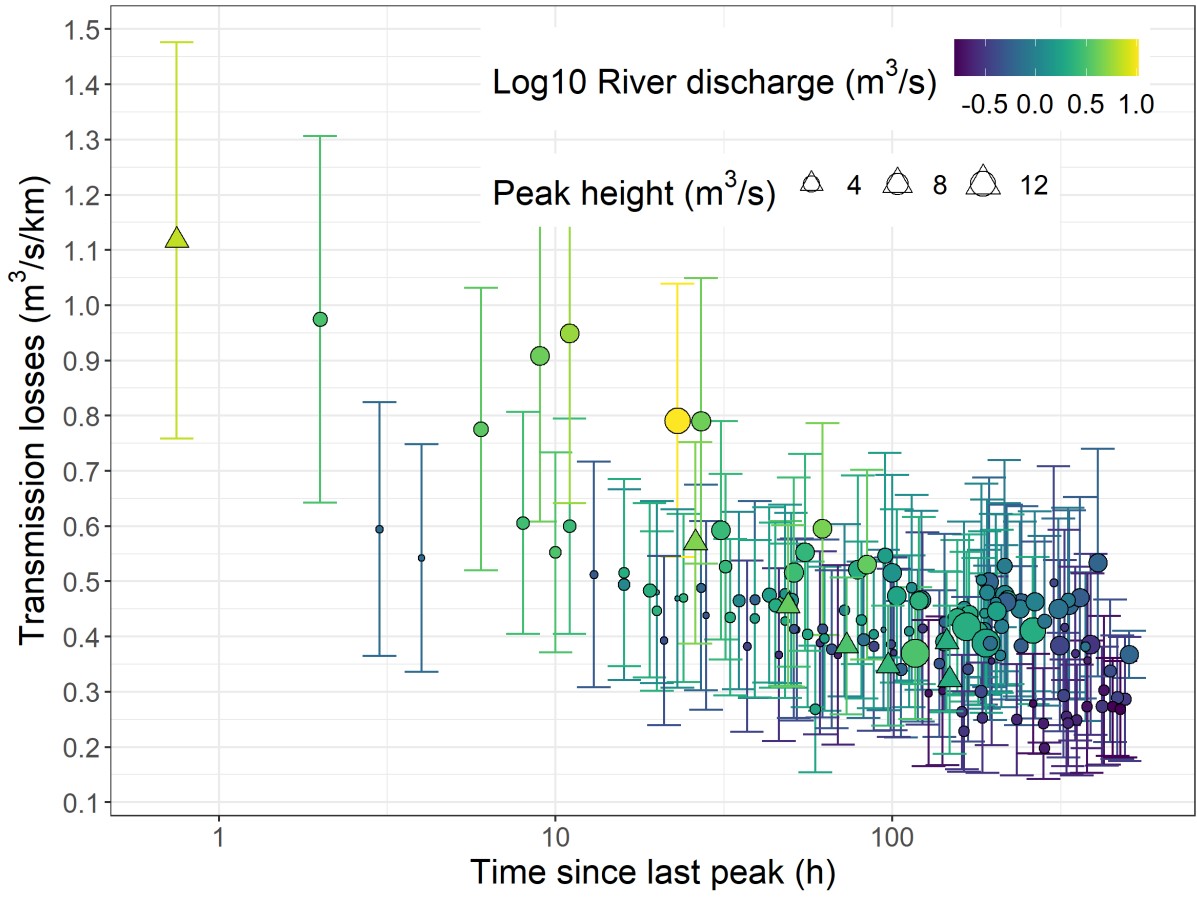

**Figure 7.** Transmission losses as a function of the time since the last peak. The colour scale represents the river discharge (log10 transformed) and the point size scale represents the peak height. Triangles indicate the September 2020 event presented in Figure 3 and circles the other data points.

### 4.3 Reconstructed transmission loss time series

The time series reconstructed using the random forest models is presented in Figure 8 and the estimated and reconstructed

duration curves in Figure 9. The random forest models managed to reproduce most of the features observed in the estimated transmission loss dataset and the associated uncertainties. The reconstructed transmission losses range between 0.16 and $1.35\,\mathrm{m^3/s/km}$ with a time average value of $0.43\,\mathrm{m^3/s/km}$. This is slightly narrower than the estimated range (0.14 to $1.5\,\mathrm{m^3/s/km}$) but with a similar time average value. Evaluating the performance of our reconstruction on the different estimated points in time, it appears that our ensemble average values corresponds well with our estimated average values

with an RMSE of $0.03\,\mathrm{m^3/s/km}$ and an NRMSE of 12%. Analysing the performance of our random forest realizations separately, the average RMSE calculated on our ensemble of random forest model fits is $0.07\,\mathrm{m^3/s/km}$ on the whole datasets,





and $0.12\,\mathrm{m^3/s/km}$ on the evaluation datasets. This corresponds to an average NRMSE of 17 and 28%, respectively. The reconstructed duration curve indicates that 56% of the studied year, the Selwyn River transmission losses downstream of our flow gauging station were between $0.25$ and $0.65\,\mathrm{m^3/s/km}$.





**Figure 8.** Transmission loss time series reconstructed (cyan) using the random forest models trained on the transmission loss data points estimated (orange) using field data and satellite images.





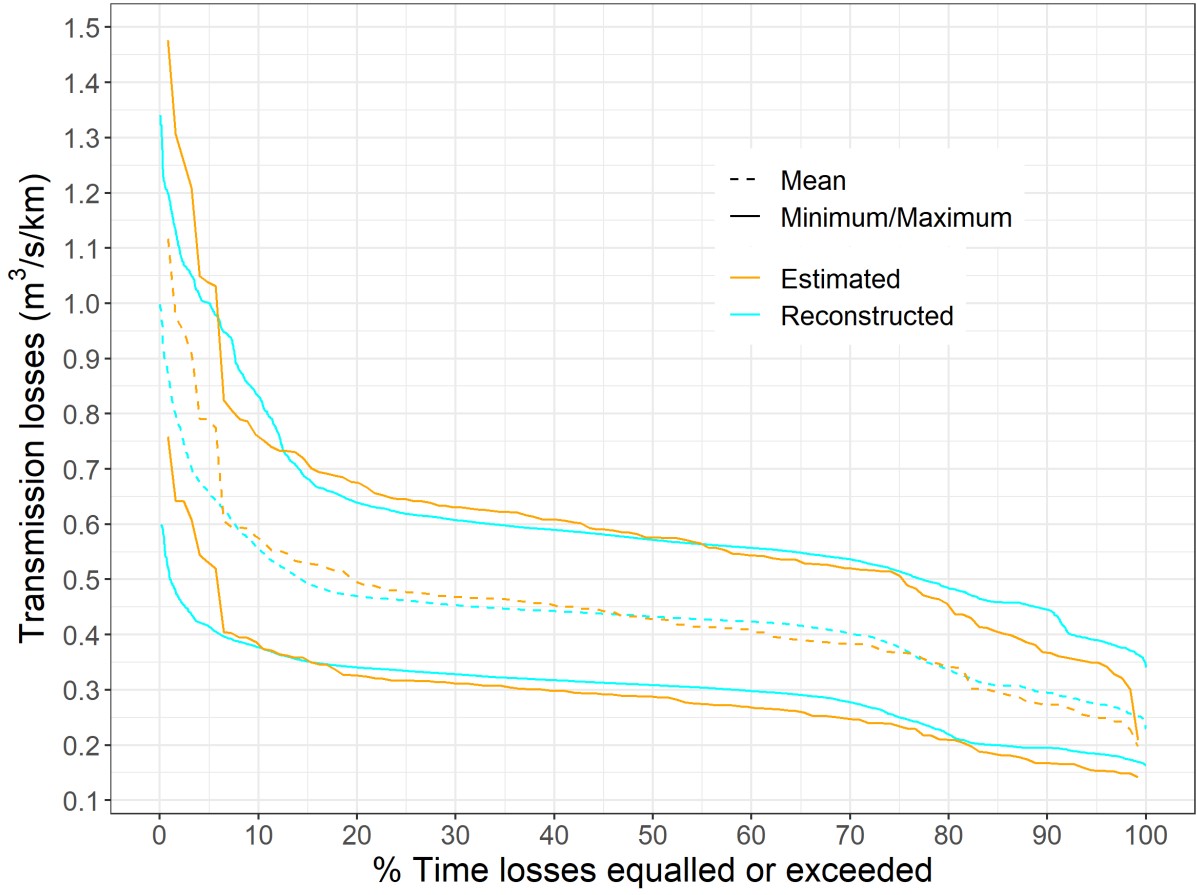

**Figure 9.** Estimated (empirical distribution, orange) and reconstructed (simulated distribution, cyan) transmission loss duration curves.

**5 Discussion**

**5.1 Distributed groundwater recharge versus local storage replenishment**

We have shown in section 4.1 and 4.2 that the transmission losses in the Selwyn River relate differently to the river stage and flow depending if the river is in a drying or in a wetting phase (few first days after peak flow). The different processes being lumped in the transmission losses can explain these contrasting behaviours. Transmission losses consist generally of

evapotranspiration and groundwater recharge. Given the sparse vegetation and the relatively high transmission losses in our study site, most of the water is expected to be lost to the groundwater, although we did not conduct a formal estimation of the respective contributions. In the remainder of this section, we assume that the estimated transmission losses represent the groundwater recharge and neglect other natural or artificial gains and losses. Furthermore, we hypothesize that the river is losing water to the groundwater in two different modes, depending whether the river is in a wetting or drying phase:



– During drying phases: the river loses water to the underlying aquifer all along its wetted length, depending on local hydraulic, geomorphological and geological properties.

– During wetting phases: the river still loses water to the underlying aquifer as during drying phases but additionally the advancing wetting front is refilling the local storage in the vicinity of the river bed. This explains the highest losses estimated shortly after peak flow, during wetting phases.

The transmission losses estimated using the method presented in this study are an average along the wetted river length. During drying phases, the wetted river length is linearly correlated to the river discharge (Figure 10). This suggests that the recharge to the aquifer is rather constant along the studied reach. Furthermore, this justifies the comparison between the transmission losses derived from the differential gauging and from the river drying front locations, although they represent losses at different scales. However, during wetting phases, a considerable amount of water is lost at the wetting front and

therefore the losses are not equally distributed along the river reach. As a consequence, the highest losses are not representative of the spatially distributed recharge to the aquifer and their values in terms of $\mathrm{m^3/s/km}$ should be interpreted with caution.

Applying our framework to the Selwyn River improved our understanding of the interactions between surface water and groundwater in this particular system. However, many unknowns remain, including the quantity of water lost at the wetting front to local storage during wetting phases. This quantity should depend on the volume of aquifer to wet and its porosity.

The regional water table under the Selwyn River at the study reach is rather deep (> $15\,\mathrm{m}$ deep, Vincent, 2005) and the river water recharging this deep aquifer is thought to flow through a variably saturated zone (Larned et al., 2011, 2008). Therefore, a significant volume of water could be lost at the wetting front to refill the local storage when the river is advancing. An ongoing research project aims at clarifying how the Selwyn is interacting with the underlying unsaturated zone and regional aquifer. The two modes of groundwater recharge identified in the Selwyn could also occur in other ephemeral river systems. Applying

the framework presented in this article to other systems could help to understand them better.



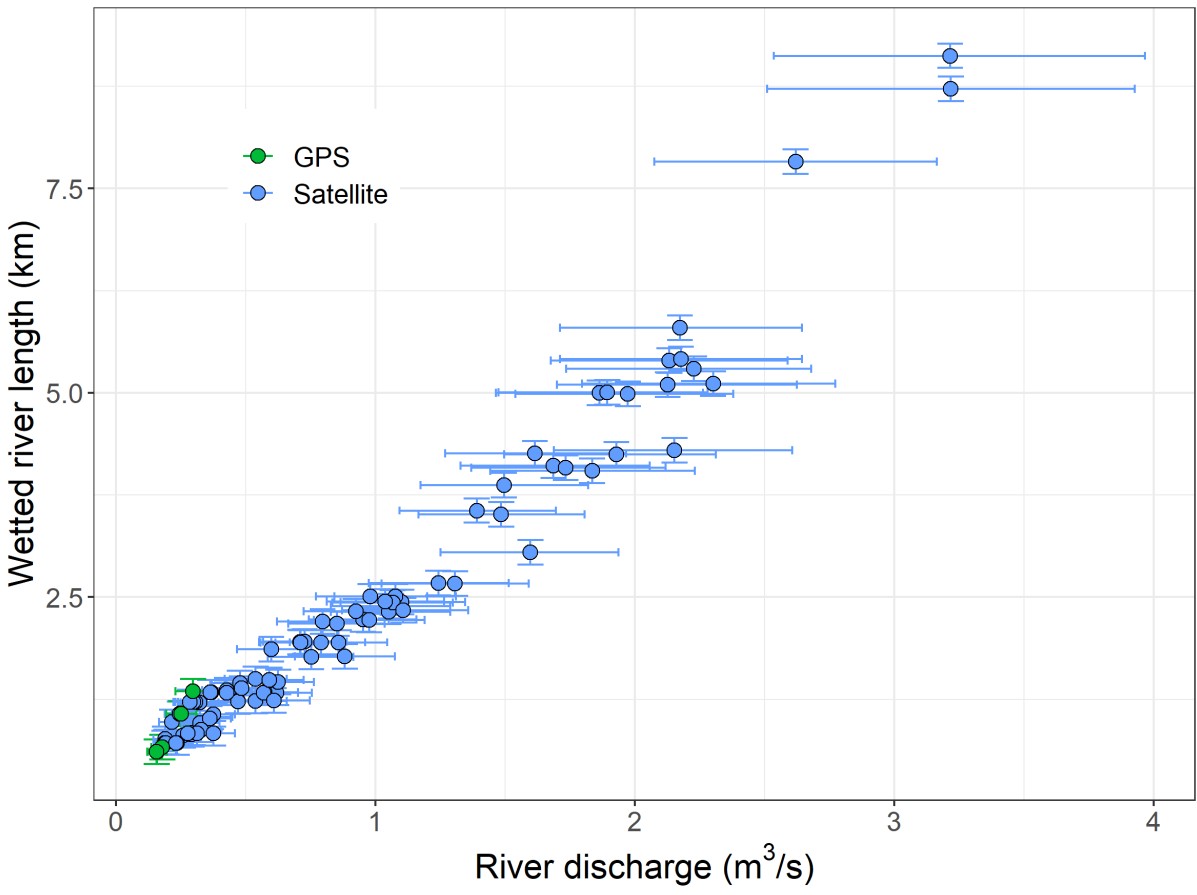

**Figure 10.** Wetted river length as a function of the river discharge, only shown for data points collected during river drying phases (more than 60 hours after a peak flow).

## 5.2 Comparison with previous studies conducted on the Selwyn River

Rupp et al. (2008) estimated transmission losses along the Selwyn River by performing river gauging manually at 18 cross-sections on a limited number of days (4 to 60, depending on the cross-section) between October 2003 and January 2007. The average transmission losses that they have estimated between the cross-sections downstream of the gauging station used in our study (i.e., Scotts Road, Figure 1) were mostly between $0.2$ and $0.5\,\mathrm{m^3/s/km}$. This is in the lower range of our base flow estimates. A more detailed comparison is difficult as our estimates differ by their spatial and temporal extent.

In a series of articles (Larned et al., 2011, 2010, 2008; Rupp et al., 2008), the ELFMOD model has been used to reconstruct the flow along the Selwyn river. Another output of the ELFMOD model is the flow permanence along the river, which was estimated to be between 20 and 75% in one of the driest reaches of the river, around 10 km downstream of Scotts Road (our gauging station). In this regard, our results differ significantly, our reconstructed wetted river length extends to the Hororata





River confluence (15 km downstream of our gauging station, where the Selwyn River is gaining water again) only during peak flows (Figure 11). Our reconstructed flow permanence curve (Figure 12) indicates that the river was dry more than 90% of the time 10 km downstream of Scotts Road during our study period. This discrepancy appears as well in the dataset used to train the models. Among the 152 drying front locations that we have identified on the satellite images and on the field between April

2020 and May 2021, no image shows the river flowing continuously to the Hororata River confluence. On the other hand the data reported by Rupp et al. (2008), collected on 118 days between October 2003 and January 2007, show that when the river flow at Coes Ford (50 km downstream our gauging station) was greater than twice the median, the entire Selwyn River was flowing.

The different results could be explained by the different approaches employed but more likely by the hydrological variability

between the study periods. The period between March 2020 and May 2021 was particularly dry in the Canterbury region (NIWA, 2021, 2020). Furthermore, a longer-term trend of decreasing low flow and wetted river length of the Selwyn River has been highlighted by McKerchar and Schmidt (2007) and Rupp et al. (2008) for the period between 1984 and 2006. More research would be needed to investigate how the recent period studied in our work (2020-2021) falls within this longer-term trend. Long-term flow record is available since 1964 at the Whitecliffs site (10 km upstream of Scotts Road) and could be used

together with satellite images to investigate the Selwyn River transmission loss inter-annual variations. The Planet monitoring library (Planet Team, 2017) used in the present study is only available from 2009 onwards but other resources might be used to cover a longer time frame, although the resolution and frequency of available images in the more distant past will be lower. Moreover, the transmission loss estimates between Whitecliffs and Scotts Road would be more difficult to interpret because they would also include a constrained and a gaining reach and thus a large spatial variability of transmission losses along the

extended reach.





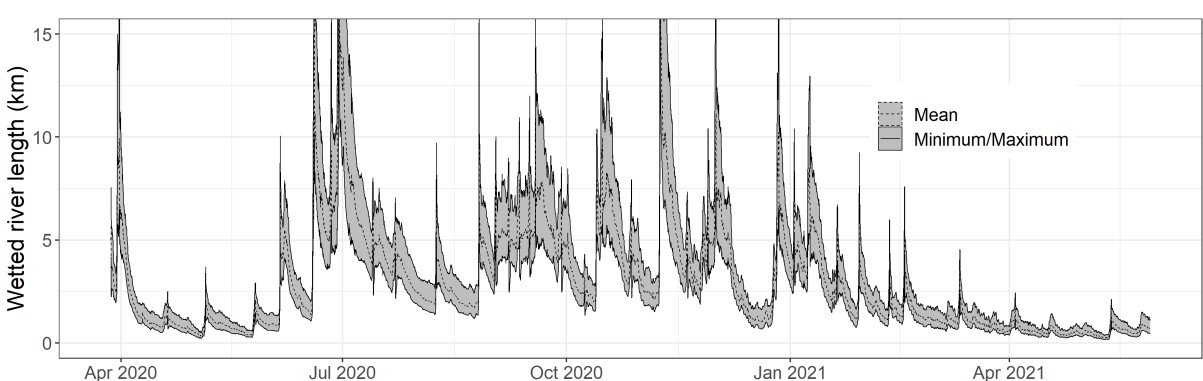

**Figure 11.** Wetted river length time series reconstructed using the random forest models.



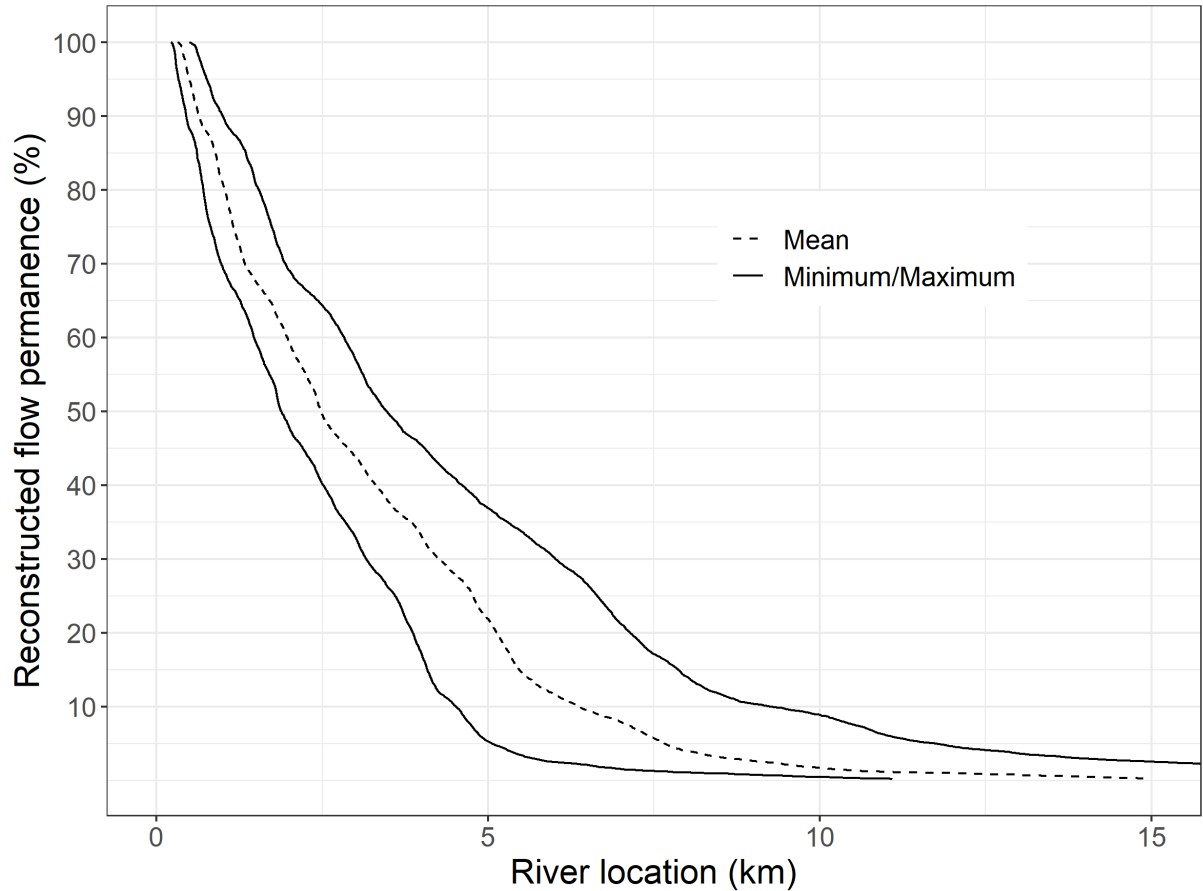

**Figure 12.** Longitudinal variation in flow permanence (proportion of year with flowing water) downstream of our gauging station (Scotts Road = 0 km).

## 5.3 Uncertainties sources and propagation

In this study, we have carried out a comprehensive assessment of the different sources of uncertainty affecting our transmission loss estimation and reconstruction. Concerning the transmission loss estimates made using the satellite images, the uncertainties range from 30 to 55%. On the one hand, the uncertainties on the river discharge derived from the rating curve represent around 20% (Appendix A). On the other hand, the uncertainties on the river drying front locations and wetted river lengths represent 10 to 30%, with increasing contribution for smaller wetted river length. The estimation made using the field GPS points are less uncertain as the river drying front location was virtually exact. As a result, the uncertainties are around 30% of the estimated values, around 20% coming from the river discharge and 10% from the wetted river lengths. For both methods, the uncertainties due to the river stage measurements are relatively low, below 4%.



Regarding the transmission loss estimates derived from the differential gauging campaigns, the uncertainties vary between
5 and 45%, depending on the measurement uncertainties (between 2.7 and 6.3%) and the ratio between the transmission losses
and the river discharge (Appendix C). At low flow, the differences between individual flow measurements (i.e. transmission
losses) are large compared to the measurement uncertainties, which lead to relatively small uncertainties on the transmission
loss estimates. However, at high flow, the differences between individual flow measurements are small compared to the
340   measurement uncertainties and therefore the resulting uncertainties on the transmission loss estimates are high. Overall, we can
state that quantifying the transmission losses from satellite imagery at our study site is not introducing much more uncertainty
than using the traditional method of differential flow gauging.

Considering all our estimates used to train the random forest regressors, the propagated measurements uncertainties show
an average value of 35%. This is higher than the normalized root mean square of the random forest fitting errors (NRMSE)
345   calculated on the whole datasets (from 11 to 27%) and in the range of the NRMSE calculated on the test datasets (from
16 to 51%). Moreover, the uncertainties on our estimated values are larger than the NRMSE calculated by comparing the
average predicted and average estimated values (12%). Therefore, we can state that our random forest ensemble is reproducing
satisfactorily our transmission loss estimates, considering the measurements uncertainties.

## 5.4   Advantages and limitations of our approach and ways forward

Quantification of the transmission losses using the framework described in this article has many advantages over traditional
methods but is also limited by our ability to identify the drying fronts on the satellite images and to reconstruct the time series
from the obtained data points.

The main advantage of our method is the reduced amount of field work needed to produce high time resolution transmission
loss estimates. Our framework only requires the installation and maintenance of a flow gauging station, which is common on
many rivers. Another requirement is the availability of clear satellite imagery with a resolution higher than the river width.
In our study, we used the Planet Monitoring collection (Planet Team, 2017), which is freely available to university-affiliated
student and researchers through their Education and Research (E&R) Program. The 3.7 m resolution of these images was just
enough to identify the river drying fronts, as the Selwyn river width is often less than 10 m. To apply the same approach to
smaller rivers, other satellite resources exist (Maxar Team, 2022; Planet Team, 2017) and pre-processing of the satellite imagery
could help (Callo, 2022). However, the time gap between two high-resolution images from other libraries is longer than the time
gap between images from the Planet Monitoring collection. Another issue with the use of satellite images would be the presence
of dense riparian vegetation or clouds, which could hinder our ability to identify the drying front on the images. In particular,
clouds tend to obscure satellite images during higher flows, as they tend to occur during or shortly after rainfall events. However
in the future, we expect that more high resolution and frequency satellite images will be available to researchers. This should
make the approach presented in this article more attractive and feasible, even for smaller rivers. Furthermore, several algorithms
have been developed to identify automatically water-covered areas from satellite images (Feyisa et al., 2014; Munasinghe
et al., 2018; Sagin et al., 2015). The difficulties described previously might complicate their utilization for our purpose, but an
investigation of the possibilities could be beneficial to future applications, especially for longer time series.



Using random forest regressors enabled us to reconstruct well the hourly transmission loss time series and their uncertainties
without requiring much effort and computational resources. Thanks to our processing of the hydrograph to calculate the time
since the last peak and the peak height for each transmission loss estimates, we could reconstruct the transmission loss time
series only using the river stage and flow time series. However, an important shortcoming of our reconstruction is that the
predicted transmission losses during the highest flow peaks (end of June and early November 2020) are not higher than the
predicted losses during lower peak flow events. This is due to the lack of data immediately after ($< 23\,\mathrm{h}$) these highest peaks.
The random forest models are then unable to extrapolate prediction outside of the conditions they have been trained on. Many
other kinds of statistical models and machine learning algorithms exist and have been applied in hydrology (Shen et al., 2021;
Solomatine and Ostfeld, 2008). Although some other machine learning algorithms could have some advantages over random
forests, the issue with extrapolation is inherent to this kind of model, which lack a representation of hydrological processes.
They are therefore unlikely to give robust prediction of the response variable outside of the training conditions (e.g. for future
scenarios simulation). A more robust alternative could lie in physically based models. One of the main motivations behind this
work is to use our estimated and reconstructed time series to evaluate different physically based models, which can then be
used for simulation of future scenarios.

## 6   Conclusions

We presented a framework to estimate the transmission losses in ephemeral rivers from satellite imagery and reconstruct their
hourly time series using random forest models. This framework was successfully applied to the Selwyn River (Canterbury,
New Zealand) for the period between March 2020 and May 2021. It is an efficient approach to quantify transmission losses
in ephemeral rivers. The method has the advantage of requiring less fieldwork and generating more data than traditional
methods like differential flow gauging, at a similar accuracy. Our results show that the transmission losses in the Selwyn River
downstream our gauging station were between $0.25$ and $0.65\,\mathrm{m^3/s/km}$ during most of the study period. However, shortly after
peak flow, when the river was advancing and wetting the surrounding sediments (i.e. wetting phases), the losses could reach
up to $1.5\,\mathrm{m^3/s/km}$. This compares quite well with previous estimates made by Rupp et al. (2008). However, we observed
and predicted a much dryer river than reported in other studies (Larned et al., 2011, 2010, 2008; Rupp et al., 2008). This is
probably due to our study period being dryer but it is unclear how this relates to decadal trends. Furthermore, studying the
relationship between the transmission losses and the river stage and discharge enabled us to improve our understanding of the
Selwyn River interactions with groundwater. We believe that the generated transmission loss time series provide a valuable
dataset to support further research efforts, especially the development of physically based models. Moreover, the presented
framework has the potential to help water management in this catchment and beyond by providing an approach to simulate the
transmission losses, groundwater recharge and wetted river length. Our framework is easily transferable to other ephemeral
rivers and can be applied to longer time series. This could provide important information at relatively low cost.





. Data are available on request from the authors, except satellite images that are own by Planet Labs.

## Appendix A:  Rating curve

The stage-discharge rating curve developed using 14 flow gauging and five inflection points is shown in Figure A1.





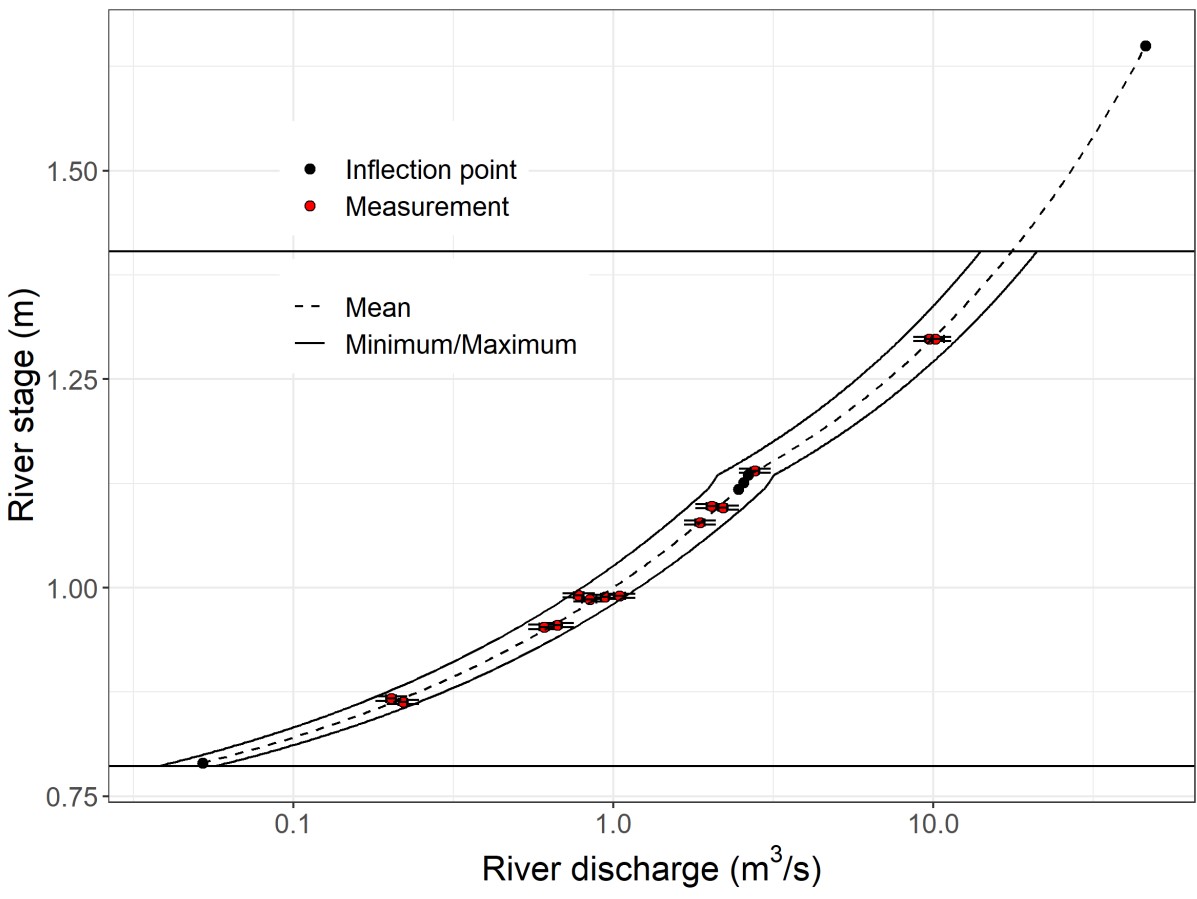

**Figure A1.** Stage-discharge rating curve (discharge was log10 transformed). The horizontal lines represent the range of river stage monitored during the study period.

## Appendix B: Hydrograph and selected flow peaks

The hydrograph and the flow peaks selected to calculate the time since the last peak and the peak height associated with each transmission loss estimate are presented in Figure B1.





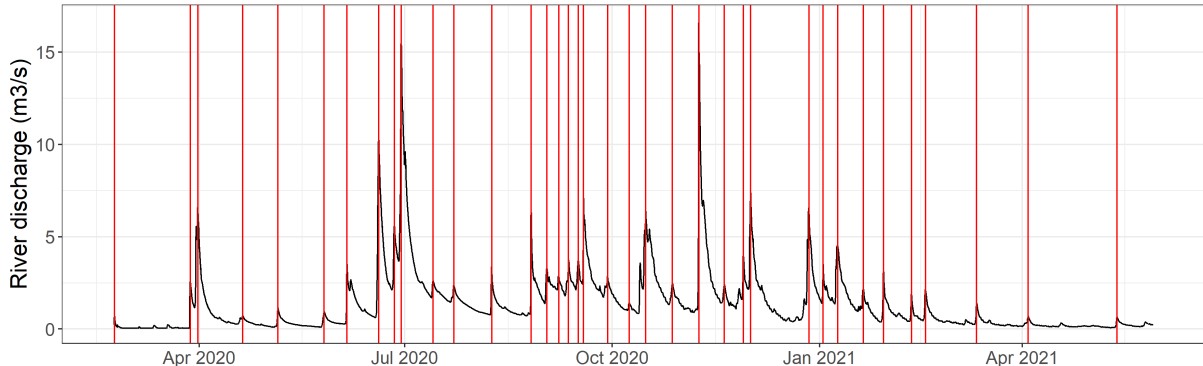

**Figure B1.** Hydrograph and flow peaks selected to calculate the time since the last peak and the peak height associated with each transmission loss estimate.

## Appendix C: Transmission losses estimation from differential gauging

The flow gauging measurements, their uncertainties and the linear model ensembles used to calculate the transmission losses are shown in Figure C1.



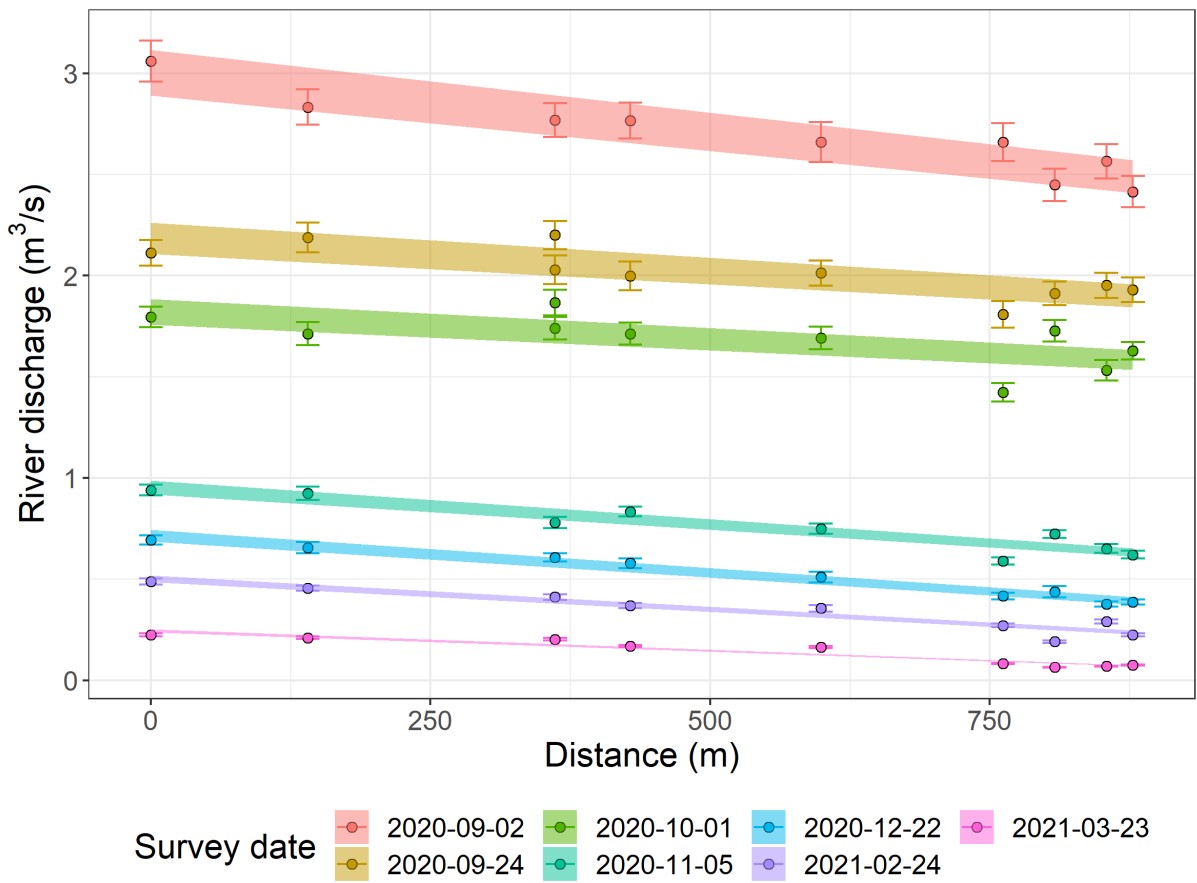

**Figure C1.** Differential flow gauging measurements and linear model ensembles used to calculate the transmission losses.

. **Author contribution**

ADC and SW conceptualized the study and developped the methodology. ADC, SW and JK collected the data. ADC analyzed the data, developed the modelling and visualization scripts. ADC wrote the manuscript and all authors reviewed it. SW and TW acquired the funding and guided the research.

. **Competing interest**

The authors declare no competing interests.



. **Acknowledgements**

We would like to thank the New Zealand Ministry of Business, Innovation and Employment for funding this research through the project "Subsurface processes in braided rivers - hyporheic exchange and leakage to groundwater" (contract LVLX1901). We are also grateful to Planet Labs for granting us access to part of their satellite imagery collection. Finally, we thank Aaron Dutton who helped with the data collection on the field.





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
