# Peer review of "Deriving transmission losses in ephemeral rivers using satellite imagery and machine learning"

_EGUsphere, 2022_

## Author Comment (AC1)

Dear Reviewer,

Thank you for the comprehensive and constructive review of our article. We take note of your comments on the need to improve the clarity, discussion and recommendations for further work. We are willing to address these comments and improve the quality of the manuscript in a revised version, if the editor request us to do so. Please find below some answers to your questions and explanations on how we would address your comments. We have also provided some information about the project within which this study was conducted, including the groundwater monitoring part of the program. Our answers are in blue and your comments are in black.

The abstract does not indicate how transmission losses can be derived from a wetted river length. An additional sentence is needed for clarification.

Thank you for pointing that out, we will add a sentence in the revised version.

l110 'inland plains' is unclear. Is coastal plains what is meant, or is there some differentiation intended between an inland plain and a coastal plain? If so, some explanation is needed.

Yes, there is a distinction usually made in this region, mainly from a geological point of view. The inland plains represent the apex of the alluvial fan and are dominated by glacial and periglacial outwash, while the coastal plains are dominated by post-glacial alluvium and marine sediments. This leads to different aquifer characteristics within these formations, with a better defined vertical series of gravel aquifers, separated by clay and peat aquitards in the coastal plains (Larned et al., 2008; Taylor et al., 1989). We will explain that in the revised version.

l171 explain what is the difference between the river bed and the active river channel – how are these identified?

We used 'river bed' to refer to the gravel bed of the river and 'active river channel' to refer to the part of the river bed where the water is currently flowing. Examples of the wetted river length following the active river channel, the river bed and the braidplain are presented below in Figure 1. We could include this figure in the revised version, maybe as an appendix.

[Figure]

*Figure 1: Wetted river length following the river braid plain, river bed and active river channel as considered in the study. The satellite image was taken on January 27, 2021, and the river drying front identified for this day is indicated on the image.*

l181 the linear model fitted in App C masks some interesting aspects of the data, which need discussion. For example there are segments that show both strong gains at some times and strong losses at others. What are possible explanations and how do these effects reflect on the very simple assumption of a linear model? We need some process insights here.

Yes, there are some aspects of these data that we did not discuss in the submitted version of the article. The small-scale (between individual gauging) variability is due to complex interactions of surface flow with the shallow (perched) aquifer. The linear models were used to estimate average transmission losses over three riffle-pool sequences and remove the localized loss/gain variability. Thus, the average loss value can be directly compared with the transmission losses obtained from the satellite image approach, as shown by the results of the study. In the revised version of the manuscript, we will provide a cross-section presenting our perceptual model of the river-aquifers system showing the shallow (perched) aquifer (as requested by Reviewer 2) and discuss how this conceptualization can explain the data.

The small-scale interactions between the river and the shallow (perched) aquifer are currently investigated using their temperature signals and will be the subject of another future publication. Briefly, an important process revealed by this analysis is that some preferential flow pathways can be activated at high flow.

L186 what is meant by the transmission loss time series? Given the spatial complexity and the multiple measurements, this phrase is ambiguous without further explanation.

We meant the time series of reach-average transmission losses for the wetted reach downstream of the flow gauging station. The dataset used to train the random forests included the transmission losses derived from the satellite images, the field GPS points and the differential gauging. We will clarify that in the text.

l215 The 'estimated transmission losses vary in time' is unclear. They also vary in space as well as time, so some clarification is needed.

Yes, this is unclear at this stage of the text, we will remove this sentence as this is explained in detail further down.

l224 need to explain where the peak flow that is referred to occurred – presumably this is at the permanent gauging station (clearly a) peak flow is very different when downstream points near the wetting front are considered, and b) there is transmission time for peak flow to propagate downstream)

Yes, we meant the peak flow at the permanent flow gauging station. This will be clarified.

l227 'transmission losses were maximum' is unclear. I assume that what is meant is 'transmission losses estimated using the modelled relationship with flow at the gauging station'. Above, this was described as estimated, but not here?

Yes exactly, we should have use 'estimated' here too. This will be added.

l232 fig 4 caption needs some qualification. These are estimated transmission losses based on the stage at the gauged hydrograph location. (similar comment for Fig 6 caption)

Yes, we will correct that and use 'estimated' consistently throughout the article, including in the figure captions.

Fig C1 shows some sections (below 750m) change from losing to gaining – so complex surface water groundwater interactions

Please see answer to comment on l181.

l314 how could hydrological variability be expected to affect the results? Presumably this relates to groundwater effects? Some thought/discussion is needed, perhaps linking to the observed variability in response shown in App C.

Yes, the groundwater storage in the shallow (perched) aquifer can explain the impact of the hydrological variability on the results. In a dry year, the shallow aquifer will be depleted and not be able to sustain the river flow as much as in a wetter year. We will add some discussion on that in the revised version and cross-sections presenting our perceptual model of the river-aquifer(s) system (as requested by Reviewer 2).

l395 – concluding comments – some thought should be given as to how to further develop insights into the response of this system. It seems to be crying out for some basic monitoring of groundwater. Is there really no data and no monitoring planned?

As mentioned briefly before, this study is part of a larger project aiming at understanding the interactions between braided rivers and groundwater (https://www.researchgate.net/project/Subsurface-Processes-in-Braided-Rivers ). Within this project, piezometers have been installed to monitor the pressure and temperature in the shallow and deeper aquifers. We are planning more publications that will focus on these data and the modelling of this river-aquifers system. However, we could include the average and the range of observed values from a few piezometers in this article to support our perceptual model, if this would improve the clarity of the manuscript.

The spelling, wording and typo errors will be corrected in the revised version as you suggested.

Antoine Di Ciacca (on behalf of the co-authors)

**References**

Larned, S.T., Hicks, D.M., Schmidt, J., Davey, A.J.H., Dey, K., Scarsbrook, M., Arscott, D.B., Woods, R.A., 2008. The Selwyn River of New Zealand: a benchmark system for alluvial plain rivers. River Res. Appl. 24, 1–21. https://doi.org/https://doi.org/10.1002/rra.1054

Taylor, C.B., Wilson, D.D., Brown, L.J., Stewart, M.K., Burden, R.J., Brailsford, G.W., 1989. Sources and flow of north Canterbury plains groundwater, New Zealand. J. Hydrol. 106, 311–340. https://doi.org/https://doi.org/10.1016/0022-1694(89)90078-4

---

## Author Comment (AC2)

Dear Reviewer,

Thank you for the comprehensive and constructive review of our article. We understand that
we should improve some aspects of this article and are willing to address most of your
comments in a revised version, if the editor request us to do so. Please find below some
answers to your questions and explanations on how we would address your comments. Our
answers are in blue and your comments are in black.

Major comments:

1. Page 2, Line 42-44: "Furthermore, as noted by Cook (2015), it is unusual for two
   gauging stations to be located on the same river without intervening tributaries.
   Therefore, quantifying transmission losses from two existing gauging stations is
   rarely possible." I disagree. In reality, for large dryland rivers, where the most studies
   on channel transmission losses were undertaken, upstream river discharge is much
   larger than runoff produced between the streamgauges. Considering allogeneic
   dryland rivers, this runoff is practically null. Therefore, quantifying transmission
   losses from two or more existing gauging stations is perfectly possible in drylands.

Thank you for pointing that out, we will remove this sentence.

2. The described perceptual model of the surface-groundwater interaction of the study
   site (2 Study site) must be much better spatially presented, showing profiles along and
   across the main river and aquifer units.

We agree that this should be better presented in our paper and will make changes in this
direction, including adding profiles (cross-sections) of the river and aquifer units, as
suggested. Please also note that Larned et al. (2008) devoted a complete article to the
presentation of the Selwyn River climate, geology, hydrology and geomorphology so we will
try to keep it rather short in our article and refer to their work for further information.

3. As you described in the study site, is it correct to say that the water lost in the
   ephemeral losing reach is immediately available again downstream?

The water lost in the ephemeral losing reach is flowing underground to and then within the
deeper regional aquifers before reaching the downstream gaining reach. The water in the
gaining reach might also partly come from other sources (e.g. Waimakariri River, land
surface recharge). Therefore the water lost in the ephemeral reach is not immediately
available in the river downstream. This will be clarified by the profiles that will be added
following your suggestions and by adding some text as well.

4. The ephemeral reach is always a losing one? Even during high floods?

We believe that the ephemeral losing reach is always losing overall, even during high flood,
because of the flat topography, high soil permeability (gravel) and the absence of tributary
around this section of the river. We will clarify that in the text if needed.

5. Page 6, Line 137: Why did you use five inflection points for the rating curve?

The rating curve is derived with concurrent discharge and stage data collected under various flow conditions. When plotting the stage and discharge data on a logarithmic scale, the correlation between discharge and stage becomes linear. Therefore, the rating curve (line) can be defined by inflection points.

The inflection points indicate where the stage-flow relationship changes within the cross-section. Therefore, the number and location of these inflection points of the rating curve depend on the channel slope, cross-sectional geometry, roughness of bed and bank materials, vegetation etc. In the case of the Selwyn River at our study site, there is one notable cross sectional widening above 1.12 m that caused a change in the linear correlation. An inflection point was drawn to indicate the rating change due to a cross-section widening above 1.12 m. The specific software we use for rating curve development requires an endpoint, a breakpoint, and a start point to change the slope. As a result, three points were drawn despite only one break of slope, which makes 5 inflection points in total.

This appears to be a bit of an over parameterization and 3 inflection points (so 1 change of slope) would have been enough to describe our data. If relevant, we can recalculate our results using only 3 inflection points in the revised version. This will change the results only very marginally because the rating curve will be virtually the same.

The details of the inflection points are not directly related to the uncertainties of the rating curve, rather the rating uncertainties are related to the gauged flow and discharge data. The sentence on page 6, line 137 might have added confusion.

In our text, we will revise the section that discusses the rating curve and the associated uncertainties to improve its clarity.

6. 3.2.1 Transmission losses derived from the river drying front locations: the main problem of this methodology is there are just five days of comparison between the GPS and satellite drying front positions, although daily satellite images were available. Therefore, more fieldwork should be done, in order to properly estimate the uncertainty on the satellite wetted river length estimation.

The method used to identify the river drying front locations on the satellite images is rather straightforward for the Selwyn River. The river drying front is simply identified visually on aerial photographs (see example below in Figure 1). Therefore, it seems to us that 5 days of validation is enough. Moreover, depending where the river fronts is located, it can be difficult to access. In this regard, we were lucky to have a dry season in summer 2020-2021, this allowed us to verify on the field the location of the drying front close to our study site. This would be much more difficult to do at the moment with the wetter weather. Five verification points might not seem much but we think that this is already a valuable and not easy to gather dataset. Furthermore, we adopted a rather conservative uncertainty of 100 m on the river drying front locations identified on satellite images.

[Figure]

*Figure 1: River drying front location identified on the satellite image taken on January 27, 2021.*

7. Page 7, Line 180: you wrote that "the higher uncertainties are typically associated with shallow and low flow in the smaller braids." However, your fitted linear model showed a rather different result with higher uncertainties related to larger flows.

We meant the relative uncertainties, which are mentioned in the previous sentence. They represent the uncertainties on the flow measurements and are in general higher for lower flows. The linear models uncertainties, which represent the uncertainties on the transmission losses are indeed increasing with flow as discussed in section 5.3.

8. Page 13, Line 245: "... and the effect of the peaks becomes an important control" How?

Because of the amount of water lost at the drying front as discussed in section. 5.1. We will make clearer the link between this section of the text and the discussion in section 5.1.

9. 4.3 Reconstructed transmission loss time series: What did we learn about the transmission losses in the Selwyn River when the machine learning approach was applied? If there is nothing to add to our understanding of the process, I suggest either excluding it or to use another time series model.

The purpose of the machine learning 'reconstruction' was not to learn more about the processes but to produce a continuous hourly record. This record can be used first to investigate the exceedance probabilities and draw the duration curve as done in this study. Second, it is useful for further research work because it provides a continuous time series that can be used to evaluate physically-based models (work in progress). Third, there is some interest in predicting continuous records of both transmission losses and active river length for water management in this catchment. This is likely to be the case in other catchments as well. Therefore, we believe that this is an important part of the framework and that the random forest regressors are doing a good job at reproducing the estimations and propagating the uncertainties. We will add these explanations to the revised version of the manuscript.

10. You should compare your study with previous studies conducted on other ephemeral streams, including those from other climates. It is fundamental to place your findings in the context of transmission loss research.

Yes, we agree that not enough effort were made to compare our study with previous studies on other ephemeral streams in the submitted version of the manuscript. We will improve that in the revised version.

Minor comments:

1. I suggest moving the Figures A1, B1 and C1 from Appendix to the main text.

Yes, we will add these Figures to the main text.

2. Page 7, Lines 190-192: "In the course of the model development, more predictors (e.g. river flow, water temperature, groundwater level, date) have been tested but they appeared to not improve significantly the predictions." Have you tried any statistical criterion, such as AIC?

We used the RMSE to evaluate the different models as explain in section 3.3.

3. Please, reconsider the terminology of "reconstructed" transmission losses, because reconstruction of time series is a quite different topic. You should use just "predicted" transmission losses.

We will change to "predicted".

4. Page 12, Lines 235-236: "The estimated transmission losses range from 0.14 to 1.5m3/s/km. Most of the estimated losses (56%) are below 0.6m3/s/km and correspond mainly to baseflow periods and river drying phases." Please, provide a box-plot of the transmission losses, and add more statistical details.

We will add a better statistical description of the dataset, including the distribution or a box plot.

5. Conclusions: It is not necessary to use citations in the conclusion.

We will remove the citations in the conclusion.

Antoine Di Ciacca (on behalf of the co-authors)

**References**

Larned, S.T., Hicks, D.M., Schmidt, J., Davey, A.J.H., Dey, K., Scarsbrook, M., Arscott, D.B., Woods, R.A., 2008. The Selwyn River of New Zealand: a benchmark system for alluvial plain rivers. River Res. Appl. 24, 1–21. https://doi.org/https://doi.org/10.1002/rra.1054

---

## Author Response (AR1)

**Reviewer 1**

Dear Reviewer,

Thank you for the comprehensive and constructive review of our article. We agree with your comments on the need to improve the clarity, discussion and recommendations for further work. We have addressed these comments in a revised version enclosed. Please find below the responses to your comments and questions and a list of the changes made in the manuscript. Your comments are in black and our answers are in blue. The modified text of the manuscript is in italic and the line numbers indicated refer to the revised version of the manuscript.

The abstract does not indicate how transmission losses can be derived from a wetted river length. An additional sentence is needed for clarification.

Thank you for pointing that out, we have added a sentence in the abstract:

Line 5: *'The transmission losses are then calculated as the flow gauged at the upstream location divided by the wetted river length.'*

Line 18 typo McMahon (also line 50)

Corrected (Line 19 and 50)

l46 downstream of

Corrected (Line 45)

l94/95 similar to

Corrected (Line 94)

l106 flows for

Corrected (Line 106)

l110 'inland plains' is unclear. Is coastal plains what is meant, or is there some differentiation intended between an inland plain and a coastal plain? If so, some explanation is needed.

Yes, there is a distinction usually made in this region, mainly from a geological point of view. The inland plains represent the apex of the alluvial fans and are dominated by glacial and periglacial outwash, while the coastal plains are dominated by post-glacial alluvium and marine sediments. This leads to different aquifer characteristics within these formations, with a better defined vertical series of gravel aquifers, separated by clay and peat aquitards in the coastal plains (Larned et al., 2008; Taylor et al., 1989). We have added these explanations in the text:

Line 110: *'When the Selwyn River reaches the alluvial plains, it first arrives in the inland plains, which are formed by the apex of the alluvial fan and are dominated by glacial and periglacial outwash.'*

Line 114: *'Further downstream, the Selwyn River reaches the coastal plains, which are dominated by post-glacial alluvium and marine sediments, and gains water from groundwater seepage.'*

Fig 1 typo 'losing'

Corrected

l143 derivative

Corrected (Line 159)

l148 discussion about time before peak unclear at this point

Moved to Line 275

l152 similar to that adopted....

Corrected (Line 166)

l153 for clarity, recall what Walters did – i.e. use the volume gauged upstream to estimate the lost discharge

Added, Line 167: *'who identified the length of the wetted reach downstream of a flow gauging station on five satellite images and calculated the transmission losses by dividing the river flow at the gauging station by the wetted river length.'*

l171 explain what is the difference between the river bed and the active river channel – how are these identified?

We used 'river bed' to refer to the gravel bed of the river and 'active river channel' to refer to the part of the river bed where the water is flowing at low flow. This has been clarified in the text and examples of the wetted river length following the active river channel, the river bed and the braidplain have been added in appendix.

Modified in the text, Line 185: *'The wetted river length can differ depending if the river active channel (where the water is flowing at low flow), the gravel riverbed or the braid plain is followed. The different lengths determined on the January 27, 2021 image are shown in Appendix A, as an example.'*

Added in Appendix A, Line 461: *'The wetted river length following the active river channel, the river bed and the braid plain are presented in Figure A1, using the satellite image taken on January 27, 2021 as an example.'*

[Figure]

*Figure A1: Wetted river lengths following the active river channel, the river bed and the braid plain as considered in the study. The satellite image was taken on January 27, 2021, and the river drying front identified for this day is indicated on the image. Image credit to Planet Team (2017).*

l181 the linear model fitted in App C masks some interesting aspects of the data, which need discussion. For example there are segments that show both strong gains at some times and strong losses at others. What are possible explanations and how do these effects reflect on the very simple assumption of a linear model? We need some process insights here.

We have now added some discussion here and we have also added a better description of our perceptual model of the groundwater – surface water interactions at our study site, including two schematic cross-sections, in section 2 (as requested by reviewer 2). These cross-sections show the shallow perched aquifer associated with the river (referred to as 'braidplain aquifer') and the deeper aquifers. However, please note that the small-scale variability of water exchanges between the river and its braidplain aquifer is beyond the scope of this article. It is currently investigated using the temperature signals measured in the river and some piezometers and will be the subject of another future publication. Briefly, an important process revealed by this analysis is that some preferential flow pathways can be activated at higher flows. This can explain the variability observed in the data.

Added, Line 203: *'The small-scale (between individual gauging) variability is due to complex interactions between the river and the braidplain aquifer. The linear models were used to estimate the reach-average transmission losses over three riffle-pool sequences and remove the localized loss/gain variability. Thus, the loss values derived from the linear models can be directly compared to the losses estimated using the satellite imagery approach. The description and explanation of this small-scale variability is beyond the scope of this study and has already be partly addressed by Banks et al. (2022). More comprehensive investigations will be the focus of future works.'*

l186 what is meant by the transmission loss time series? Given the spatial complexity and the multiple measurements, this phrase is ambiguous without further explanation.

Explanation added, Line 210: *'Random forest regression models were trained on a dataset including the estimates obtained from the satellite images, the field GPS points and the differential gauging surveys. These models enable us to predict the hourly reach-average transmission losses for the wetted reach downstream of the flow gauging station on the days and times without measurements. This provides us with a continuous hourly transmission losses time series covering the entire study period.'*

l215 The 'estimated transmission losses vary in time' is unclear. They also vary in space as well as time, so some clarification is needed.

Yes, this is unclear at this stage of the text, this sentence has been removed as this is explained in detail further down.

Paragraph modified, Line 241: *'In this section, we first explain how the reach-average transmission losses downstream of our gauging station vary in time for one particular event in September 2020, then show the complete dataset of estimated values and lastly present our predicted time series.'*

l224 need to explain where the peak flow that is referred to occurred – presumably this is at the permanent gauging station (clearly a) peak flow is very different when downstream points near the wetting front are considered, and b) there is transmission time for peak flow to propagate downstream)

Yes, we meant the peak flow at the permanent flow gauging station.

Added, Line 250: *'at the permanent gauging station'*

l227 'transmission losses were maximum' is unclear. I assume that what is meant is 'transmission losses estimated using the modelled relationship with flow at the gauging station'. Above, this was described as estimated, but not here?

Yes, clarified in the text, Line 266: *'estimated using the satellite images and the rated flow at the gauging station'*

l232 fig 4 caption needs some qualification. These are estimated transmission losses based on the stage at the gauged hydrograph location. (similar comment for Fig 6 caption)

We have clarified that in the caption of Figure 7 (formerly Figure 3), Figure 8 (formerly Figure 4), Figure 10 (formerly Figure 6) and Figure 11 (formerly Figure 7).

Added, Figure 7 caption: *'estimated using differential gauging ('Gauging', red) and river drying front locations identified on satellite imagery ('Satellite', blue)'*

Added, Figure 8 caption: *'estimated using differential gauging ('Gauging', red) and river drying front locations identified on satellite imagery ('Satellite', blue)'*

Added, Figure 10 caption: *'using differential gauging, field GPS points and satellite images'*

Added, Figure 11 caption: *'Estimated transmission losses using differential gauging, field GPS points and satellite images'*

Fig C1 shows some sections (below 750m) change from losing to gaining – so complex surface water groundwater interactions

Please see response to comment on Line 181.

l301 differ in

Corrected (Line 352)

l314 how could hydrological variability be expected to affect the results? Presumably this relates to groundwater effects? Some thought/discussion is needed, perhaps linking to the observed variability in response shown in App C.

Yes, the groundwater storage in the shallow perched (braidplain) aquifer can explain the impact of the hydrological variability on the results. In a dry year, the braidplain aquifer would be depleted and not able to sustain the river flow as much as in a wetter year. We have added some discussion in section 5.2 and two cross-sections presenting our perceptual model of the river-aquifers system in section 2 (as requested by Reviewer 2).

Added in section 5.2, Line 367: '*This led to low water level and storage in the braidplain aquifer [@Banks2022] and limited the ability of this shallow aquifer to sustain the river flow as much as in a wetter year.*'

l395 – concluding comments – some thought should be given as to how to further develop insights into the response of this system. It seems to be crying out for some basic monitoring of groundwater. Is there really no data and no monitoring planned?

As mentioned briefly before, this study is part of a larger project aiming at understanding the interactions between braided rivers and groundwater (https://www.researchgate.net/project/Subsurface-Processes-in-Braided-Rivers ). Within this project, piezometers have been installed to monitor the water level and temperature in the shallow (braidplain) and deeper aquifers. We are planning more publications that will focus on these data and the modelling of this river-aquifers system.

In the revised version of this article, we have added a section 'Future work' after the section 'Conclusions', Line 448: *'Some aspects of the groundwater – surface water interactions at our study site still need to be investigated in more details. On the one hand, there is evidence of complex interactions, variable in space and time, between the Selwyn River and its braidplain aquifer. On the other hand, the infiltration from the braidplain to the deeper aquifer might be a simpler process, as suggested by the relatively stable losses estimated during drying phases in this study. Further research is needed to understand better these processes, their spatio-temporal variability and how they can be appropriately simulated. This is the focus of an ongoing research programme within which piezometers have been installed to monitor the water level and temperature in the shallow (braidplain) and deeper aquifers. In addition, active distributed temperature sensing surveys are being carried out to assess the small-scale variability of groundwater – surface water interactions [@Banks2022]. Furthermore, we are developing physically-based models of various complexities to represent the river-aquifers system and enable us to get further insights in the system response and to simulate future scenarios.'*

l411 developed

Corrected (Line 464 and 465)

Best regards,

Antoine Di Ciacca (on behalf of the co-authors)

**Reviewer 2**

Dear Reviewer,

Thank you for the comprehensive and constructive review of our article. We agree with most of your comments on the need to improve some aspects of this article. We have addressed these comments in a revised version enclosed. Please find below the responses to your comments and questions and a list of the changes made in the manuscript. Your comments are in black and our answers are in blue. The modified text of the manuscript is in italic and the line numbers indicated refer to the revised version of the manuscript.

Major comments:

1. Page 2, Line 42-44: "Furthermore, as noted by Cook (2015), it is unusual for two gauging stations to be located on the same river without intervening tributaries. Therefore, quantifying transmission losses from two existing gauging stations is rarely possible." I disagree. In reality, for large dryland rivers, where the most studies on channel transmission losses were undertaken, upstream river discharge is much larger than runoff produced between the streamgauges. Considering allogeneic dryland rivers, this runoff is practically null. Therefore, quantifying transmission losses from two or more existing gauging stations is perfectly possible in drylands.

These two sentences have been removed from the text.

2. The described perceptual model of the surface-groundwater interaction of the study site (2 Study site) must be much better spatially presented, showing profiles along and across the main river and aquifer units.

We agree that this should be better presented in our paper. Two cross-sections (profiles), one along and one across the river and the main aquifers have been added in Figure 2. The text presenting the study site (section 2) has been extended to better explain our perceptual model of the groundwater – surface water interactions. Moreover, the section 5.1 has been modified in different places to include this conceptualization.

[revised manuscript text omitted]

3. As you described in the study site, is it correct to say that the water lost in the ephemeral losing reach is immediately available again downstream?

The water lost in the ephemeral losing reach is flowing underground to and then within the deeper regional aquifers before reaching the downstream gaining reach. The water in the gaining reach also partly come from land surface recharge. Therefore the water lost in the ephemeral reach is not immediately available in the river downstream.

Added, Line 117: *'The lag time analysis performed by Larned et al. (2008) suggests that it takes several weeks for the water to infiltrate from the upstream gaining river section to a deeper aquifer (~20m deep). Part of this water might be captured by the downstream gaining section of the river after travelling underground in a complex network of aquifers.'*

4. The ephemeral reach is always a losing one? Even during high floods?

Yes the ephemeral losing reach and its braidplain aquifer are always losing water to the deeper aquifer overall, even during high flood, because of the flat topography, high soil permeability (gravel), the absence of tributary along this section of the river and the deep aquifer water table.

Added Line 132: *'However, the studied reach and its braidplain aquifer are always loosing water to the deeper aquifer overall, even during high floods. Surface runoffs are limited by the flat topography, high soil permeability (gravels) and absence of tributary along this section of the river. The deeper aquifer water table is much lower (~15m) than the river and its braidplain aquifer.'*

5. Page 6, Line 137: Why did you use five inflection points for the rating curve?

The rating curve is derived with concurrent discharge and stage data collected under various flow conditions. When plotting the stage and discharge data on a logarithmic scale, the correlation between discharge and stage becomes linear. Therefore, the rating curve (line) can be defined by inflection points.

The inflection points indicate where the stage-flow relationship changes within the cross-section. Therefore, the number and location of these inflection points of the rating curve depend on the channel slope, cross-sectional geometry, roughness of bed and bank materials, vegetation etc. In the case of the Selwyn River at our study site, there is one notable cross sectional widening above 1.12 m that caused a change in the correlation. An inflection point was drawn to indicate the rating change due to a cross-section widening above 1.12 m. The specific software we use for rating curve development requires an endpoint, a breakpoint, and a start point to change the slope. As a result, three points were drawn despite only one break of slope, which makes 5 inflection points in total.

This appears to be a bit of an over parameterization and 3 inflection points (so 1 change of slope) would have been enough to describe our data. In the revised version, we have recalculated our results using only 3 inflection points. This will change the results only very marginally because the new rating curve is virtually the same than the old one.

The details of the inflection points are not directly related to the uncertainties of the rating curve, rather the rating uncertainties are related to the gauged flow and discharge data. The sentence on page 6, line 137 might have added confusion.

In our text, the section that discusses the rating curve and the associated uncertainties has been revised to improve its clarity, Line 151: *'At the cross section where the stage was recorded, there is one notable widening above 1.12m that caused a change in the correlation between stage and discharge, therefore we introduced one break of slope in our rating curve.*

*The uncertainties of the manual gauging data varied from 2.4 to 6.5%. The fitting errors between our manual flow measurements and the rating curve ranged from 0 to 15%, with an average of 5 and a standard deviation of 7%. Considering these two sources of uncertainties, we assumed 20% of uncertainty on the rated river flows.'*

6. 3.2.1 Transmission losses derived from the river drying front locations: the main problem of this methodology is there are just five days of comparison between the GPS and satellite drying front positions, although daily satellite images were available. Therefore, more fieldwork should be done, in order to properly estimate the uncertainty on the satellite wetted river length estimation.

We found another validation data point collected on January 21, 2021 by another institution during our study period. This has been added to our dataset and is shown on Figure 9 and following figures. This data point shows again that the uncertainties that we have considered on the location of the drying front identified on satellite images are large enough and that the transmission losses calculated from the satellite images and from the GPS field data are similar.

We would also like to highlight that the method used to identify the river drying front locations on the satellite images is rather straightforward for the Selwyn River. The river drying front is simply identified visually on aerial photographs (see example below in Figure 2). Therefore, we think that six days of validation is enough. Moreover, depending where the river front is located, it can be difficult to access. In this regard, we were lucky to have a dry summer 2020-2021 because this allowed us to verify on the field the location of the drying front close to our study site. This would be much more difficult to do at the moment with the wetter weather. Six verification points might not seem much but we think that this is already a valuable and not easy to gather dataset. Please note that although satellite images are taken every day, they were usable only around every 3 days during our study period (147 out of 427 days). Furthermore, we adopted a rather conservative uncertainty of 100 m on the river drying front locations identified on satellite images.

The text has been modified as we have now six drying front locations from field GPS points instead of five.

Line 8: *'six occasions'*

Figure 1 caption: *'153 drying fronts'*

Line 179: *'six different days'*

Line 180: *'153 drying fronts'*

Line 345: *153 drying front locations*

[Figure]

*Figure 2: River drying front location identified on the satellite image taken on January 27, 2021. Image credit to Planet Team (2017).*

7. Page 7, Line 180: you wrote that "the higher uncertainties are typically associated with shallow and low flow in the smaller braids." However, your fitted linear model showed a rather different result with higher uncertainties related to larger flows.

We meant the relative uncertainties, which are mentioned in the previous sentence. They represent the uncertainties on the flow measurements and are in general higher for lower flows. The linear model uncertainties, which represent the uncertainties on the estimated transmission losses are indeed increasing with flow as discussed in section 5.3.

Added, Line 195 and 196: *'relative'*

8. Page 13, Line 245: "... and the effect of the peaks becomes an important control"
   How?

Because of the amount of water lost at the wetting front to the braidplain aquifer as discussed in section. 5.1.

Text modified to improve the clarity, Line 278: *'At low flow (up to 1m stage and 1m3/s discharge), the relationship between the river stage and the transmission losses is relatively linear and the estimated transmission losses vary from 0.14 to 0.83m3/s/km. At higher flow (> 1m stage and 1m3/s discharge), transmission losses stop increasing linearly and reach a plateau around 0.45m3/s. As explained in section 4.1, transmission losses decrease linearly with the logarithm of the time since the last peak during wetting phases. The peak height appears to control the maximum values estimated during peak flows. Small peaks have only a minor impact, even on losses estimated shortly after peak flows. However, transmission losses estimated shortly after higher peak flows are very dependent on the time since the last peak and could reach more than 1m3/s/km in several instances (Figure 10). The relation between the transmission losses behaviour and hydrological processes is further discussed in section 5.1.'*

9. 4.3 Reconstructed transmission loss time series: What did we learn about the transmission losses in the Selwyn River when the machine learning approach was applied? If there is nothing to add to our understanding of the process, I suggest either excluding it or to use another time series model.

The purpose of the machine learning 'reconstruction' is not so much to learn more about the processes but to produce a continuous hourly record, which is useful for further work. First, this record was used within this study to investigate the exceedance probabilities and draw the duration curve. Second, it is useful for further research work because it provides a continuous time series that can be used to evaluate physically-based models (work in progress). Third, there is some interest in predicting continuous records of both transmission losses and wetted river length for water management in this catchment. This is likely to be the case in other catchments as well. Therefore, we believe that this is an important part of the framework and that the random forest regressors are doing a good job at reproducing the estimations and propagating the uncertainties. We have added these explanations to the revised version of the manuscript, in section 5.4.

Added, Line 421: *'This provides us with a continuous hourly record of transmission losses, which is particularly useful for further work. First, this record was used within this study to investigate the exceedance probabilities and draw the duration curve. Second, the continuous transmission losses record can be used to evaluate physically based models. Third, there is some interest in predicting continuous records of both transmission losses and wetted river length for water management in this catchment. This is likely to be the case in other catchments as well.'*

10. You should compare your study with previous studies conducted on other ephemeral streams, including those from other climates. It is fundamental to place your findings in the context of transmission loss research.

Yes, we agree that not enough effort were made to compare our study with previous studies on other ephemeral streams in the submitted version of the manuscript. We have added a paragraph in the section 5.2 of the revised version.

Added in section 5.2: *'Comparing our results to the dataset including 73 reaches from 31 streams sourced from different studies by McMahon and Nathan (2021) indicates that the mean reach transmission losses per event predicted for the Selwyn River (0.43GL/km) is much higher than the median of the dataset (0.046GL/km) but lower than the 90th percentile (1.1GL/km). In this regard, the Selwyn River transmission losses appear to be rather high. However, the transmission losses in the Selwyn River are still considerably lower than estimated in large ephemeral rivers under arid climate (e.g., Lange (2005) reported a mean of 6.13GL/km and Jarihani et al. (2015) a mean of 6.79GL/km GL/km). An important difference is that we have estimated the transmission losses including the water lost at the drying front. This affected our largest loss estimates and the relationship between transmission losses and river stage and discharge. The only other application of the approach followed in this study was conducted by Walter et al. (2012) on a larger river but using only five satellite images. Their estimates ranged between 0.15 and 0.25m3/s/km. This is lower than estimated in this study for the Selwyn River and could be explained by the higher sediment permeability at our study site. Unfortunately, a comparison of the time dynamics of the estimated transmission losses and their relationship with the river stage and discharge is not possible because of the limited number of data points reported by Walter et al. (2012). More studies using this approach would be needed to investigate how this varies between ephemeral river systems. The increasing availability of satellite images should make that possible in the future.'*

Minor comments:

1. I suggest moving the Figures A1, B1 and C1 from Appendix to the main text.

Done, moved to Figures 3, 4 and 5.

2. Page 7, Lines 190-192: "In the course of the model development, more predictors (e.g. river flow, water temperature, groundwater level, date) have been tested but they appeared to not improve significantly the predictions." Have you tried any statistical criterion, such as AIC?

We used the RMSE to evaluate the different models as explain in section 3.3.

Added, Line 219: *'in terms of root mean square error (RMSE)'*

3. Please, reconsider the terminology of "reconstructed" transmission losses, because reconstruction of time series is a quite different topic. You should use just "predicted" transmission losses.

Changed everywhere to 'predicted'.

4. Page 12, Lines 235-236: "The estimated transmission losses range from 0.14 to 1.5m3/s/km. Most of the estimated losses (56%) are below 0.6m3/s/km and correspond mainly to baseflow periods and river drying phases." Please, provide a box-plot of the transmission losses, and add more statistical details.

The distribution is already shown in the form of a duration (cumulative frequency) curve in Figure 13. We have added more statistical details in the text.

Added, Line 264: *'The average value of the estimated transmission losses is 0.44m3/s/km and the median is 0.43m3/s/km. The upper and lower quartiles are 0.48 and 0.37m3/s/km, respectively. A duration (cumulative frequency) curve calculated from this dataset is shown in section 4.3.'*

5. Conclusions: It is not necessary to use citations in the conclusion.

Citations removed from the conclusion

Best regards,

Antoine Di Ciacca (on behalf of the co-authors)

---

## Author Response (AR2)

Dear Editor,

Thank you for handling our manuscript. We have only added a sentence in the acknowledgements section to thank the reviewers.

Best regards,

Antoine Di Ciacca